*Resource*

EMBO
Molecular Medicine

# Spatial transcriptomics elucidates localized immune responses in atherosclerotic coronary artery

Joana Campos [1,8✉], Jack L McMurray[1,8], Michelangelo Certo [2,8], Ketaki Hardikar[1], Chris Morse[1], Clare Corfield[1], Bettina M Weigand[1], Kun Yang [3], Mohsen Shoaran [3], Thomas D Otto [3,4], Desley Neil [2,5], Pasquale Maffia [3,6,7,9] & Claudio Mauro [2,9✉]

## Abstract

Atherosclerosis is characterized by the accumulation of lipids and immune cells in the arterial wall, leading to the narrowing and stiffening of blood vessels. Innate and adaptive immunity are involved in the pathogenesis of human atherosclerosis. However, spatial organization and roles of immune cells during disease progression remain poorly understood. A better understanding of the immune response's contribution to atherosclerosis progression could unveil novel therapeutic targets to mitigate plaque development and rupture, ultimately reducing cardiovascular events burden. Here, we utilised GeoMx® and CosMx™ technologies to analyse serial sections of human coronary arteries from patients with varying degrees of atherosclerotic lesion severity. Our work comprises a series of investigations and integrates findings from both datasets, including pathway analyses, cell typing, and neighbourhood analysis. This workflow highlights the power of combining these spatial transcriptomics platforms to elucidate biological processes at the single-cell level. Our approach unbiasedly identifies molecules and pathways of relevance to support the understanding of atherosclerosis pathogenesis and assess the potential for novel therapies.

**Keywords** Atherosclerosis; Immune Cells; GeoMx; CosMx; Spatial Omics
**Subject Categories** Cardiovascular System; Chromatin, Transcription & Genomics; Methods & Resources

## Introduction

Significant mechanistic advances in cardiovascular immunology have largely been driven by technological developments. It all began with the development of monoclonal antibodies, which enabled the use of immunofluorescence to map key immune cells in human atherosclerotic plaques for the first time in the 1980s (Shoaran and Maffia, 2024; Jonasson et al, 1986, 1985). Progress continued with the first flow cytometry analysis of the mouse aorta (Galkina et al, 2006), advanced imaging of arterial lymphocyte recruitment (Maffia et al, 2007), and the development of omics techniques such as cytometry by time of flight (CyTOF) (Cole et al, 2018) and single-cell RNA sequencing (scRNA-seq) (Fernandez et al, 2019a; de Winther et al, 2023). We now understand that the immune response in atherosclerosis is complex, with various immune cells and phenotypes detectable at different stages of the disease and in different layers of the arteries.

Recently, spatial transcriptomics has been applied to assign mRNA readouts to specific locations within carotid endarterectomy sections and coronary unstable lesions (Schneider et al, 2023; Kawai et al, 2023; Ravindran et al, 2024; Sun et al, 2023; Gastanadui et al, 2024; Lai et al, 2025). However, this powerful technique has not yet been used to achieve single-cell transcriptomic analysis of coronary arteries. Moreover, the vascular samples analysed to date provide limited insight into the early stages of disease progression and lack a comprehensive view of perivascular immune mechanisms. To fully understand the immune response's significant role in the plaque, adventitia, and perivascular space (Hu et al, 2015), a comprehensive approach that includes multiple tissue layers is essential. This also involves longitudinal pathology studies, which underpin our approach to imaging coronary arteries at different stages of atherosclerosis development. Additionally, it is crucial to test various platforms, compare them directly, and integrate datasets to provide a holistic understanding.

In this study, we employed NanoString GeoMx® Digital Spatial Profiler (DSP) and CosMx™ Spatial Molecular Imager (SMI) analyses on intact human coronary arteries at various stages of disease progression to probe the immunological processes underpinning atherosclerosis. These cross-sections of human coronary arteries, affected by atherosclerosis and obtained from explanted hearts of individuals who received heart transplants, include the full thickness of the vessel wall and surrounding adipose tissue with vasa vasorum.

[1]Propath UK Limited, Hereford, UK. [2]School of Infection, Inflammation and Immunology, College of Medicine and Health, University of Birmingham, Birmingham, UK. [3]School of Infection & Immunity, College of Medical, Veterinary and Life Sciences, University of Glasgow, Glasgow, UK. [4]Laboratory of Pathogens and Host Immunity, Centre National de la Recherche Scientifique, and Institut National de la Santé et de la Recherche Médicale, Université de Montpellier, Montpellier, France. [5]Department of Cellular Pathology, University Hospitals Birmingham NHSFT, Birmingham, UK. [6]Department of Pharmacy, School of Medicine and Surgery, University of Naples Federico II, Naples, Italy. [7]Africa-Europe CoRE in Non-Communicable Diseases & Multimorbidity, African Research Universities Alliance (ARUA) & The Guild of European Research-intensive Universities, Glasgow, UK. [8]These authors contributed equally: Joana Campos, Jack L McMurray, Michelangelo Certo. [9]These authors jointly supervised this work: Pasquale Maffia, Claudio Mauro. ✉E-mail: joana.campos@propath.co.uk; c.mauro@bham.ac.uk

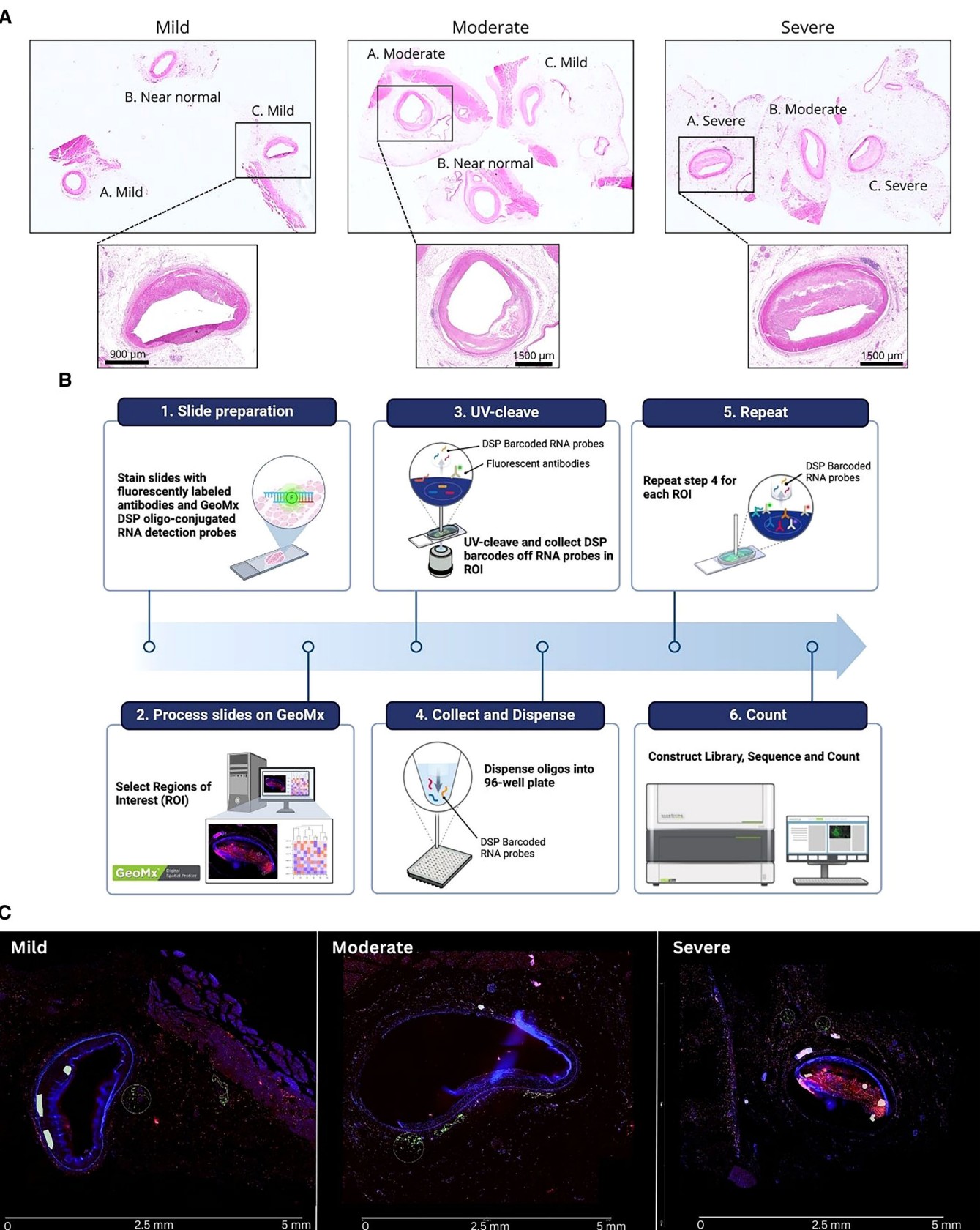

**Figure 1. Histological analysis and workflow for GeoMx®.**

(A) Representative haematoxylin and eosin (H&E) stained sections showing arterial lesions of varying severity (mild, moderate and severe). Specific regions are highlighted at a higher magnification to reveal morphological differences across lesion severity, indicating progressive changes from near normal to severe. (B) Workflow for spatial transcriptomics using GeoMx® DSP: slide preparation with morphology markers and ~18,000 oligo-conjugated RNA probes; selection of ROIs in each sample analysed; cleavage with UV light of barcodes from RNA probes; collection and release of collected barcodes onto a 96-well plate for all selected ROIs; generation of a cDNA library for next generation sequencing and upload of resulting sequencing data onto the GeoMx® DSP. (C) Fluorescent imaging of arterial lesions with varying severities, reflecting those shown at higher magnification in (A). Fluorescent imaging is a prerequisite for choosing regions of interest (ROIs) for downstream profiling by GeoMx. Samples were stained with SYTO13 (nuclear dye, blue), CD45 (pan-leucocyte marker, yellow) and CD4 (T cell subset marker, red). Representative ROIs chosen for downstream spatial profiling are indicated by white circles. Source data are available online for this figure.

Spatial transcriptomics has rapidly gained significance in many biomedical research fields, as it allows for transcriptomic analysis in a spatial cellular context. Our study is the first to report transcriptomic analysis of sequential human coronary artery sections using two combined spatial transcriptomic platforms. Using the GeoMx® DSP platform, we conducted whole transcriptome analysis (~18,000 transcripts) across regions of interest (ROIs) in coronary artery sections from three subjects with atherosclerotic lesions. ROIs covered infiltrating immune cells in the plaque and adventitia, as well as smooth muscle tissue. With the CosMx™ SMI platform, we accomplished single-cell transcriptomic analysis of the same tissues, using serial sections matching those used for GeoMx® DSP, where cells were profiled with a universal cell characterisation panel consisting of 970 RNA probes (including 20 custom probes selected based on GeoMx dataset analyses).

Here, we describe a complementary workflow for transcriptomic analysis utilising these two platforms and demonstrate how combining these analyses provides a novel and powerful tool that we anticipate will shape experimental approaches in translational and clinical medicine beyond atherosclerosis and cardiovascular disease.

# Results

## A spatial profiler approach in human atherosclerotic lesions

For this study, we accessed human coronary artery samples from subjects who underwent heart transplants. These samples contained the full circumference and thickness (all layers) of the artery, along with adjacent tissues. The samples show atherosclerosis progression from near-normal vessels (with adaptive intimal thickening but no atherosclerosis, ≤AHA type 3) to early intermediate, hereafter referred to as mild (eccentric intimal thickening with early atheroma showing foam cells/extracellular lipid), late intermediate, hereafter referred to as moderate (increased size of the atheromatous plaque with luminal encroachment, development of a cap, and formation of a plaque shoulder), and severe lesions (large atheromatous plaques with marked narrowing of the lumen and increased fibrosis within the plaque, >AHA type 3), (Fig. 1A). Sections from this cohort were hybridised with ~18,000 barcoded RNA probes, which cover the whole human transcriptome (Fig. 1B) and then stained with morphology markers (SYTO13, CD45 and CD4). Staining intensity thresholds were used to segment ROIs into CD45+CD4− and CD45+CD4+ rich segments. While these segments are not pure, they were enriched for the indicated immune cells. Barcodes were collected from a total of 55 segments throughout the available coronary arteries to sample immune cells (CD45+CD4− and CD45+CD4+) infiltrating various layers of atherosclerosis-affected vessels, i.e. the plaque, adventitia,

and smooth muscle cell layers (Fig. 1C). As controls, some CD45−CD4− segments in nerve and muscle areas of the coronary arteries were also profiled. Approximately 3000 genes passed QC and were further analysed.

## Infiltrating immune cell transcriptomic heterogeneity unveiled with the whole transcriptome atlas

As an initial approach, we performed unsupervised hierarchical clustering of immune cell segments across the plaque and adventitia (early [≤AHA type 3] and advanced [>AHA type 3] lesions), which are the main anatomical sites of immune cell infiltration in the vessel and revealed distinct transcriptomic profiles across the dataset (Fig. 2A). Comparing immune cells from different layers of the atherosclerotic vessel, we found the most differentially expressed genes in the adventitia relative to the plaque. In the adventitia, PLA2G2A, C7 and C3 were up-regulated, while SULF1, FN1 and SPP1 were more prominent in the plaque (Fig. 2B).

To further investigate these differences, we grouped the lesions into early (near normal and mild) and advanced (moderate and severe). Immune cells in the adventitia of early vs. advanced lesions displayed diversity in gene expression, with several genes significantly enriched at different stages of disease progression (Fig. 2C). APOD and ADH1B were up-regulated in early lesions, and CXCR4 and SPOCK2 in advanced lesions (Fig. 2D). Using a publicly available R package provided by NanoString (SpatialOmicsOverlay version 1.4.0), we were able to transpose gene expression levels, as exemplified by APOD and SPOCK2, onto immunofluorescence scans of mild and severe lesions, showing increased gene expression in the adventitia of advanced lesions (Fig. 2E).

To confirm the source cell types underlying the observed differential gene expression between different layers of the vessel (Fig. 2B) and across varying levels of disease severity (Fig. 2C) in the GeoMx® dataset, we performed deconvolution analysis (estimating the proportions of different immune cell types in each group or condition) utilising publicly available single-cell RNA sequencing reference data from human coronary arteries (Barcia Durán et al, 2024). As shown in Fig. 2F,G, we observe an increase in B cells and a decrease in macrophages within the adventitia compared to the plaque (F); similarly, we find a higher percentage of lymphocytes in the adventitia of moderate or severe lesions as compared to normal arteries or those with mild atherosclerosis (G).

## Whole transcriptome atlas informs pathway analysis

The GeoMx whole transcriptome atlas provides one of the most comprehensive tools for conducting pathway analysis. To investigate the enrichment of differentially expressed genes into

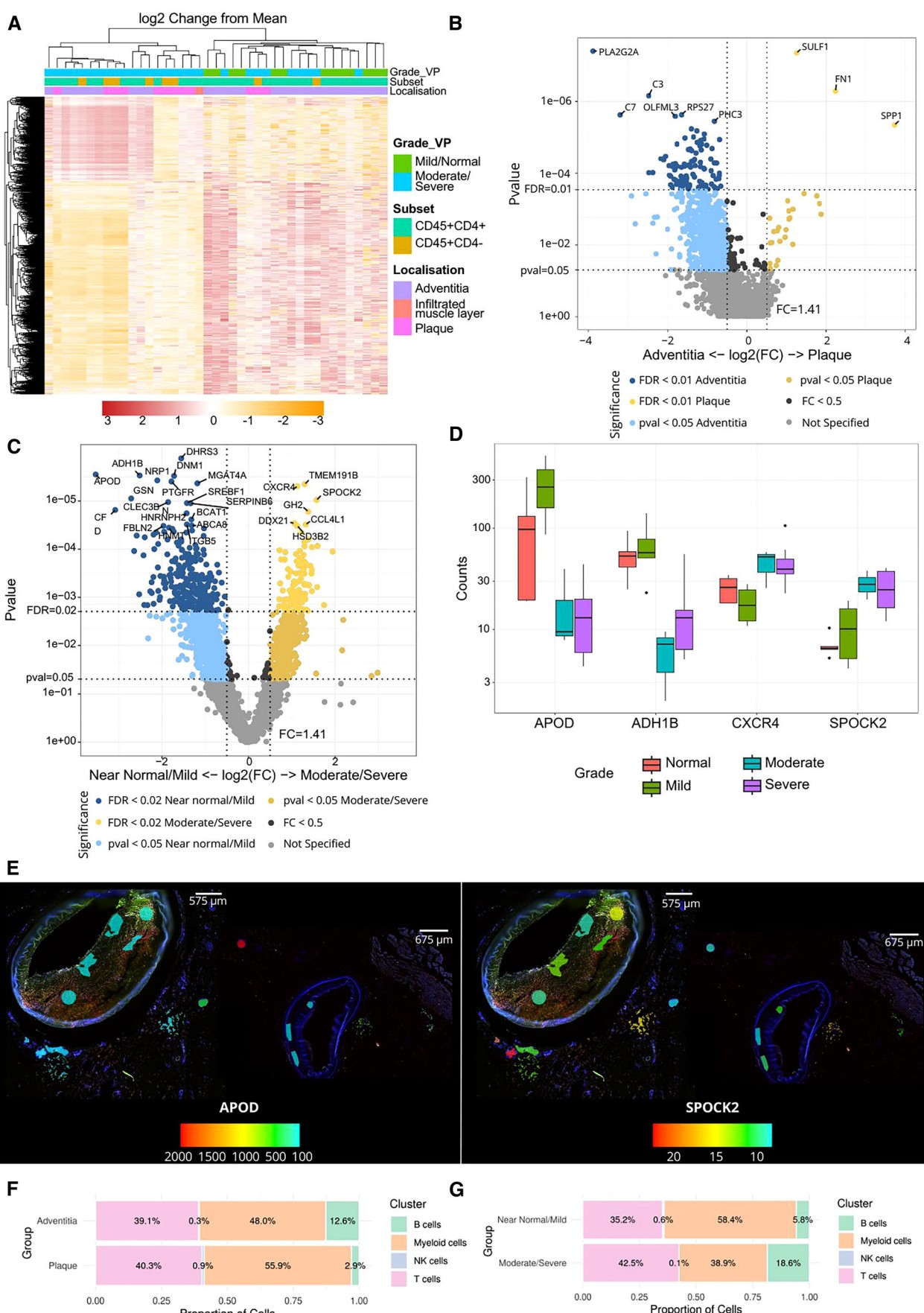

Figure 2. Spatial transcriptomic profiling of arterial lesions using GeoMx®.

(A) Heatmap demonstrating the log2 fold change from the mean of gene expression of all targets post-QC. Heatmap is annotated by lesion severity (mild/normal and moderate/severe), CD45+ subset (CD45+CD4+ and CD45+CD4−), and tissue location (adventitia, infiltrated muscle layer and plaque). The scale bar represents log2 change from the mean gene expression. (B) Volcano plot exhibiting results of differentially expressed targets between CD45+ segments located in the adventitia and in the plaque (test: linear mixed model; test correction: Benjamini–Hochberg; $n = 28$ (Adventitia), 12 (Plaque). (C) Volcano plot displaying results of differentially expressed targets between CD45+ segments in the adventitia of early lesions (near normal/mild) and those located in the adventitia of advanced lesions (moderate/severe) (test: linear mixed model; test correction: Benjamini–Hochberg); $n = 13$ (near normal/mild), 12 (moderate/severe). (D) Box plots displaying counts of the top two differentially expressed genes in early (APOD and ADH1B) vs advanced lesions (CXCR4 and SPOCK2), as shown in (C); $n = 3$ (moderate), 5 (normal), 8 (mild) and 9 (severe). Centre line, median; box edges, interquartile ranges (25th and 75th percentiles); whiskers, 1.5 times the interquartile range; outliers are shown as individual points. (E) Overlays of APOD and SPOCK2 onto the immunofluorescence scan of a severe (left) and a mild (right) lesion, highlighting the targets' expression in each profiled ROI. Overlay generated using the SpatialOmicsOverlay R package. The scale bar represents gene-normalised counts. Figure 1C was reused to create this panel. (F, G) Bar charts showing the estimated relative proportions of immune cell types in CD45-enriched (CD45+) segments located in the adventitia and in the plaque (F) or in the adventitia of early (near normal/mild) and advanced (moderate/severe) lesions (G). Source data are available online for this figure.

biologically meaningful pathways, we performed gene set enrichment analysis (GSEA) using the 'Biological Processes' section of the Gene Ontology (GO) database. Interestingly, significantly enriched pathways were identified in CD45+ cells in the adventitia of near-normal/mild vessels as compared to moderate/severe vessels (Fig. 3A; Dataset EV1), suggesting a de-differentiated/stem-like immune cell phenotype in severe lesions. Pathways related to actin filament organisation and sterol transport were among the most significantly altered (Fig. 3B). However, while the actin filament organisation pathway is overall significantly increased in early lesions, a subset of targets assigned to this pathway is up-regulated in advanced lesions, highlighting the possibility of further investigations from these pathway analysis outputs (Fig. 3C).

## Single-cell spatial transcriptomic resolution using CosMx spatial molecular imager

We extended our spatial transcriptomics workflow to analyse serial sections from the same samples used in the GeoMx® study, using NanoString's latest spatial profiler, CosMx™. This is a single-cell spatial imager with a similar slide preparation method to that required for GeoMx® studies (Fig. 4A). Oligo-conjugated RNA probes are hybridised to tissue sections on glass slides, which are then imaged within the instrument following a series of reporter cycling steps.

For this study, coronary artery sections were profiled with the CosMx™ universal cell characterisation RNA panel, which contained a core of 950 probes and an additional set of 20 custom probes (specific to this study and informed by the GeoMx® DSP study). Fields of View (FOVs, $n = 235$) were placed to select areas of interest to be imaged and profiled. In this study, FOVs were placed in such a way that the full thickness of the coronary arteries, including the plaque (when present), the media, the adventitia and infiltrating immune cells, were analysed (Fig. 4B). This represents a significant advantage of CosMx™ over GeoMx®'s limited ROI area.

The current workflow for CosMx™ studies includes an automatic data push to the AtoMx™ Spatial Informatics Platform (SIP), a NanoString cloud platform for dedicated data analysis of CosMx™-generated datasets. However, as of the writing of this article, AtoMx SIP remains a work in progress and does not yet provide a comprehensive solution for analysing complex datasets like those produced by CosMx™. Therefore, our use of AtoMx SIP in this study was limited to data upload from the CosMx™ instrument, quality control (QC), and data export to an AWS bucket. For downstream analyses, we employed publicly available R packages, resulting in a bespoke pipeline for analysing CosMx™ datasets (Fig. 4C).

One key addition to our CosMx™ data analysis pipeline is a cell- and FOV-filtering step, which excludes any cells or entire FOVs that do not meet the specified QC metrics. We believe that the data trimming resulting from this filtering step (Fig. 4D) is essential for any downstream analyses and should not be considered optional, as only data that successfully passes QC is carried forward.

## Single-cell typing provided by census

We used a census-automated cell annotation tool, which is pre-trained on the Tabula Sapiens dataset for vasculature, to perform cell typing on the coronary arteries profiled in this study. This approach led to the identification of 11 distinct cell types: B cells; CD4+ memory T cells; CD4+ helper T cells; CD8+ cytotoxic T cells; two subsets of endothelial cells; fibroblasts; macrophages; myofibroblasts; and two subsets of smooth muscle cells (Fig. 5A). In the left panel of Fig. 5A, all cells (represented as individual dots) are displayed in both their physical space within the tissue and the flow cell, while the right panel shows cells in high-dimensional UMAP space, highlighting specific cell type enrichments across lesion severity.

To assess the success of the cell-typing method, we carefully examined the immunofluorescence scans generated by CosMx™ alongside the cell-typing readout. The vessel's media layer was typed as being composed of two types of smooth muscle cells, while the plaque was mainly identified as macrophage and CD4+/CD8+T cell infiltrates (Fig. EV1A,B). A single layer of endothelial cells was also identified lining the vessel lumen, in agreement with current knowledge of these tissues (Fig. EV1A,B).

An enrichment of several cell types linked with disease progression was observed when the UMAP dataset was split by lesion severity (Fig. 5B). As expected in near-normal vessels, the most abundant cell types were endothelial cells, smooth muscle cells, fibroblasts and myofibroblasts. Amongst immune cells, a certain degree of macrophage was already present, while adaptive immune cells were underrepresented. It has to be taken into account that these are cross-sections of coronary arteries from patients with a certain degree of pathology, and hence, although phenotypically still normal, we defined them as near normal. Smooth muscle cells 1 and 2 were significantly enriched in severe lesions compared to early lesions, demonstrating the phenotypic modulation of vascular smooth muscle cells (VSMCs), which can adopt various phenotypes influencing atherosclerosis development and progression in both beneficial and detrimental ways (Basatemur et al, 2019; Kawai et al, 2023). Similarly, myofibroblasts showed a noticeable increase as the disease advanced, consistent with findings from scRNA-seq of human coronary arteries (Mosquera et al, 2023; Wirka et al, 2019). In contrast to much of the

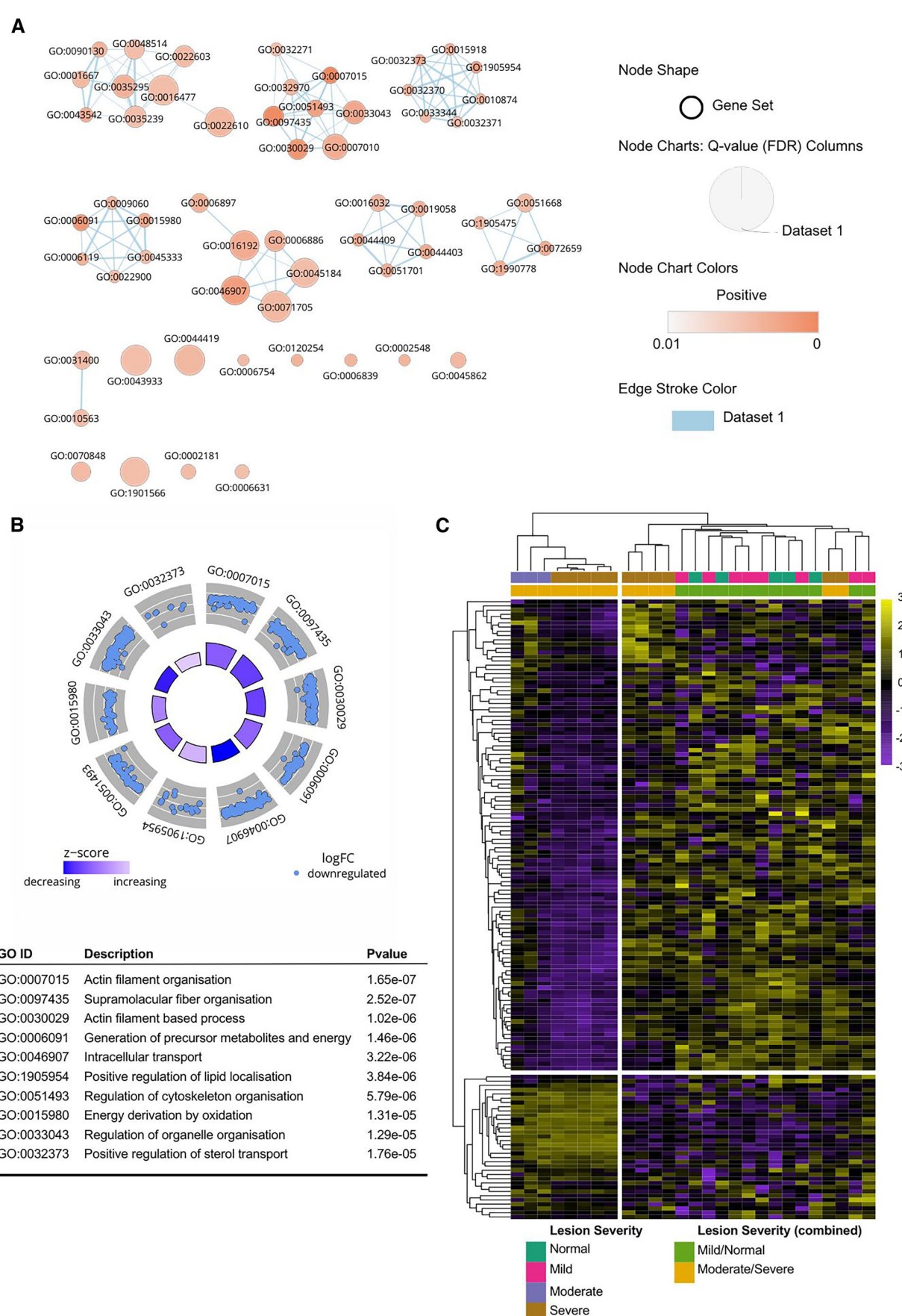

**Figure 3.  Pathway analysis from a spatially resolved GeoMx® dataset.**

(A) Gene set analysis using the Biological Processes section of the Gene Ontology database reveals 58 differentially enriched pathways in CD45+ segments in the adventitia of early compared to advanced lesions as determined by fgsea. The enrichment map was produced with Cytoscape, where the size of each circular node represents the size of an enriched pathway. Jaccard similarity scores evaluated gene overlap between pathways, whereby a thicker line represents increasing similarity. Finally, nodes were coloured by the degree of enrichment (FDR value), whereby a deeper red indicates a more significant pathway. Only gene sets with an FDR <0.01 are shown. (B) GOCircle plot representing the top ten differentially enriched pathways. Scale bar represents z-score calculated as (up-regulated genes – down-regulated genes)/√total genes, indicating over (high z-score) or underrepresentation (low z-score) of genes in a pathway. Outer circle of the plot displaying a scatter plot for each pathway of the logFC of the assigned genes (each gene represented as a dot in the scatter plot). Inner circle plotting a bar plot where bar colour corresponds to z-score (blue, decreased; white, increased; and lavender, unchanged) and bar size reflects degree of significance (log10-adjusted P value). Description and p values for the top ten enriched pathways are also displayed. GOCircle plot generated using GOplot 1.0.2 R package. (C) Heatmap displaying log2 change from the mean of targets included in the top enriched pathway (GO:0007015 Actin Filament Organisation).

traditional literature on atherosclerosis, but in line with recent studies (Fernandez et al, 2019b), CD4+ T helper/memory, CD8+, and B cells were most enriched in severe lesions compared to early ones. Furthermore, a distinct subset of endothelial cells was identified in advanced lesions, marked by an elongation of the endothelial cells from earlier lesions, which became more pronounced in moderate and severe cases. This is consistent with the emergence of a transcriptionally distinct subset of endothelial cells limited to advanced lesions (Depuydt et al, 2020) (Fig. 5B).

Furthermore, CosMx™ allows resolution of gene expression within specific cell types. Using example genes and pathways from the GeoMx® dataset (Fig. 2), we show that APOD was expressed by fibroblasts, myofibroblasts, endothelial cells, and macrophages across all severities (Fig. 5C), while SPOCK2 was primarily enriched in CD8+ and CD4+ T cells in advanced lesions (Fig. 5D). We then performed cell typing of the whole tissues for each lesion severity (Fig. 5E). The cell typing output in the representative severe lesion highlights a macrophage-rich plaque occupying a significant part of the vessel lumen, and an organised immune cell aggregate, mostly consisting of B cells and CD4+ memory/helper T cells, resembling arterial tertiary lymphoid organs (ATLOs) found in the perivascular space during experimental atherosclerosis (Hu et al, 2015). Zooming in on the ATLO associated with the severe lesion revealed that B cells, CD4+ T cells, and high endothelial venule cells, typical of ectopic lymphoid structures, were most enriched (Fig. 5E).

Finally, we plotted the same genes as in Figs. 2E, and 5C,D (APOD and SPOCK2) onto the whole tissue. As expected, APOD was mainly expressed in the media and adventitia, while SPOCK2 was primarily associated with immune cells, as evidenced by its expression in the ATLO (Fig. 5F).

## Cell quantification and neighbourhood analysis in CosMx datasets

Next, we quantitatively explored the CosMx™ dataset in terms of cellular density, gene expression, and cellular neighbourhoods. We first quantified the immune cell density across lesion severity. By normalising cell counts to the number of cells per FOV, we observed an increased immune infiltrate of all tested populations in the 'severe' category (Fig. 6A). To further probe the microenvironment within lesions, we identified ten cellular neighbourhoods based on the proximity (within 20 μm) of cell types using k-means clustering. We then examined the spatial relationships of these neighbourhoods in both physical and UMAP space (Fig. 6B). Generally, the neighbourhoods consisted of a single cell type, as highlighted by the UMAP mapping of neighbourhood annotations onto the cell type UMAP from Fig. 5A, suggesting that most cell

types reside in neighbourhoods with cells of the same annotation. To probe this further, we calculated the frequency of each neighbourhood across each of the lesion severities (Fig. 6C). While all neighbourhoods were present in all severity levels (near normal, mild, moderate, severe), their frequencies varied. Neighbourhoods 1 and 7 were particularly enriched in near normal lesions, while Neighbourhoods 3, 5, 9 and 10 were enriched in advanced lesions. Finally, we plotted a heatmap of neighbourhood compositions (Fig. 6D), showing that myofibroblasts and endothelial cells were enriched in near-normal lesions, while immune cell populations such as macrophages (Neighbourhoods 5, 3 and 10) and B cells (Neighbourhood 9) were more prominent in advanced lesions.

## Leveraging the integration of CosMx™ and GeoMx® to explore targets beyond CosMx™ coverage

When it comes to target expression, GeoMx® data is limited to a unique value per ROI/AOI within the tissue, as shown by the expression of ANXA2 mapped onto the immunofluorescence scan using SpatialOmicsOverlay (Fig. 7A). A significantly higher resolution can be achieved by CosMx™ when investigating the expression of the same target as per-cell specific expression levels are provided (Fig. 7A).

Currently, NanoString provides algorithms for spatial deconvolution of cell type estimates from GeoMx® regions. These algorithms rely on a previously generated and annotated cell type matrix to estimate cell type abundances within the ROIs (Fig. 7B). Whilst other experiments can assess the utility of these algorithms by nature of experimental design (ROIs selected for a specific cell type), we were presented with a unique opportunity in being able to directly compare all predicted cell types simultaneously by overlaying ROIs (GeoMx®) and FOVs (CosMx™) covering the same tissue areas. Furthermore, spatially deconvoluted GeoMx® data can be improved by using an input matrix relevant to the tissue of interest; therefore, we hypothesised that this spatial deconvolution may be improved if we used a matrix derived from our CosMx™ dataset (Fig. 7C).

Firstly, we demonstrate a lack of correlation between the cellular estimates derived from the inbuilt matrix and the annotated ROI positions (Fig. EV2A). Here, regions chosen based on CD45+CD4+ were largely being predicted to be populations consisting of macrophages and NK cells, rather than CD4+ populations. To ameliorate this, we used our Census-annotated CosMx™ data to produce a reference matrix (Fig. EV2B) for spatial deconvolution of GeoMx® ROIs. Whilst there seemed to demonstrate regions of gene specificity for the cell populations within the heatmap, after refining these genes to those retained post-filtering

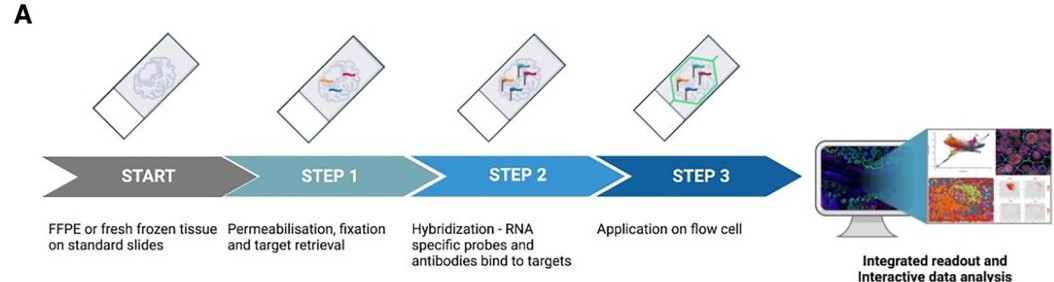

**A**

START — FFPE or fresh frozen tissue on standard slides

STEP 1 — Permeabilisation, fixation and target retrieval

STEP 2 — Hybridization - RNA specific probes and antibodies bind to targets

STEP 3 — Application on flow cell

Integrated readout and Interactive data analysis

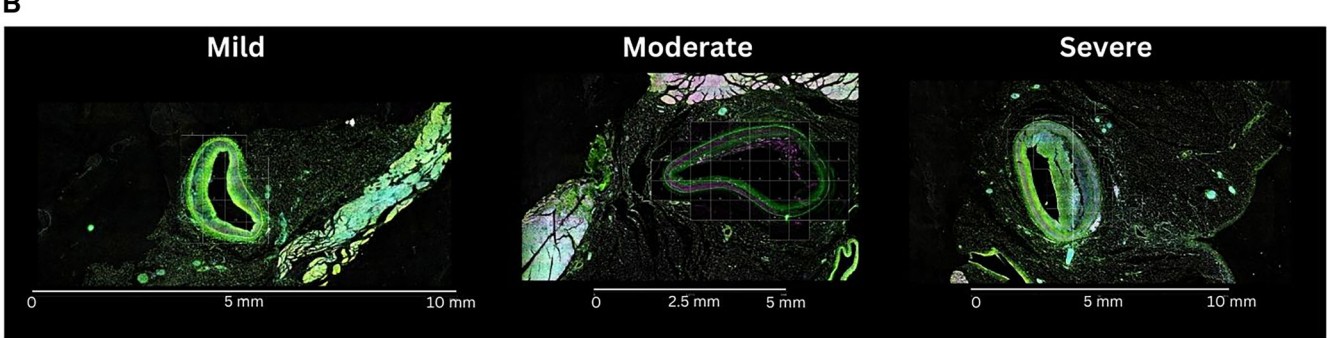

**B**

Mild | Moderate | Severe

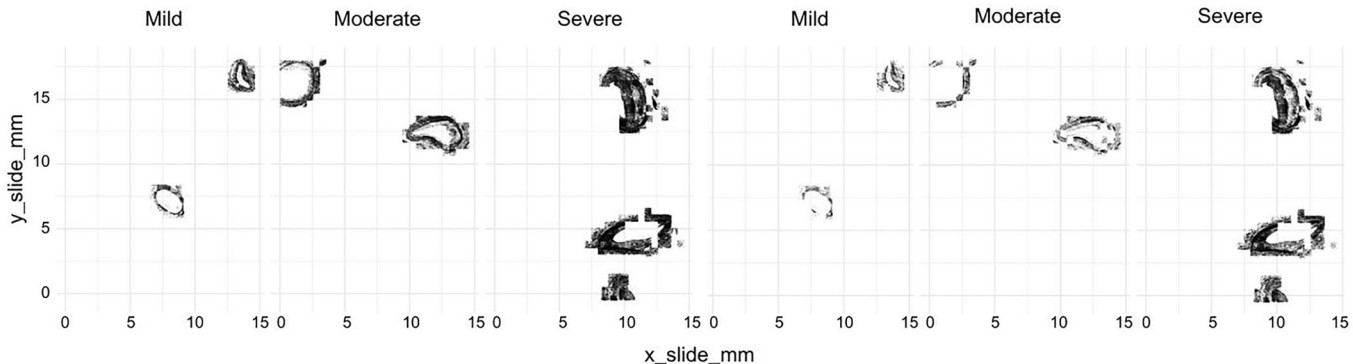

**C** Cell coordinates in XY space

Mild | Moderate | Severe | Mild | Moderate | Severe

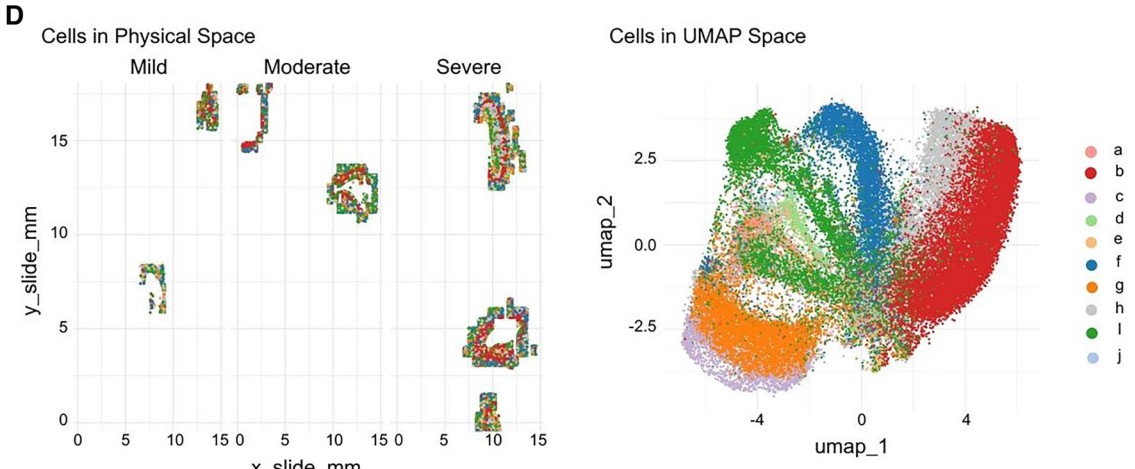

**D**

Cells in Physical Space

Mild | Moderate | Severe

Cells in UMAP Space

**Figure 4. Workflow for single-cell spatial transcriptomics using the CosMx™ SMI platform.**

(A) Schematic representation of steps involved in CosMx™ studies. (B) Immunofluorescence scans generated by CosMx of representative mild, moderate and severe lesions, depicting FOV placement. Sections were stained with DAPI (nuclear staining, not visible), B2M/CD298 (cell membrane markers, cyan), PanCK (pan-epithelial cell marker, green) and CD45 (pan-leucocyte marker, red). (C) Data analysis for CosMx™ was split between AtoMx™ and R; the diagram highlights which parts of the process were performed on which platform. (D) Plots displaying cells in the slide physical space for each lesion severity analysed. Left panel: all cells included in the dataset before the cell and FOV-filtering step. Right panel: cells remaining in the dataset post-cell and FOV filtering. Source data are available online for this figure.

in the GeoMx dataset, these regions were all but lost, particularly in the CD4+ compartment (Fig. EV2C). The limited overlap of genes between our GeoMx® and CosMx™ datasets ($n = 366$) likely hampered this attempt, and future attempts will undoubtedly improve with the release of the 6000plex CosMx™ panel. Therefore, unfortunately, spatial deconvolution using the CosMx™-derived reference matrix produced similar results to that of the inbuilt matrix (Fig. EV2D).

Lastly, we hypothesised these estimates would allow us to perform reverse spatial deconvolution, whereby cell estimates would be used as input to estimate the gene expression of relevant targets within different cell populations. Using this approach, we expected to be able to compare the expression of any given target (present in the GeoMx® whole transcriptome atlas panel but absent in CosMx™) across different lesion severities (Fig. 7C). However, unfortunately, due to the limited number of targets within the CosMx™ panel affecting the resolving of the GeoMx® data, we could not confidently investigate the cell types expressing any given target.

This workgroup aimed at using CosMx™ data to generate an annotated single-cell matrix to be used as the source of single-cell data to infer cell estimates in GeoMx® studies (Fig. 7C). However, at this time, with the resources available (i.e. panel plexity discrepancy between GeoMx® and CosMx™), this approach was not successful. Nonetheless, we believe that in the future, the ability to generate a single-cell matrix from serial sections of the same samples as those used in any given GeoMx study would be an invaluable and most accurate resource.

## Discussion

In this study, we employed a combined NanoString GeoMx®/CosMx™ approach to spatially resolve cell types and gene expression across different stages of atherosclerosis in human coronary arteries. To our knowledge, this is the first study to combine these two technologies for analysing coronary artery disease. Previously, similar combined spatial profiling approaches have been used exclusively in cancer research (Moffet et al, 2023; Derry et al, 2023).

Furthermore, the few studies that have presented spatial RNA sequencing data from cardiovascular tissues have focused mainly on human carotid endarterectomy specimens. These typically include only the luminal atherosclerotic plaque, with at most a thin strip of media, or unstable coronary artery lesions. Few of these studies utilised 10x Visium technology to uncover site-specific pathways driving human atherosclerotic plaque instability in endarterectomy samples (Sun et al, 2023; Bleckwehl et al, 2025; Lai et al, 2025; Gastanadui et al, 2024) and coronary arteries with an erosion phenotype (Kawai et al, 2023). In previous comparative analyses, the GeoMx® platform was found superior for deep molecular profiling of known regions, making it more suitable for

hypothesis-driven research compared to the 10x Visium platform (Wang et al, 2023). More recently, the Resolve platform has been used with a limited panel of genes in carotid or femoral endarterectomy samples (Ravindran et al, 2024).

Hence, our study is the first to apply a combined GeoMx®/CosMx™ approach to human coronary arteries, focusing on disease progression and immune response in the vascular environment. Additionally, because of the diverse clinical samples we utilised, it is the first to assess the full thickness of the arterial wall and surrounding adipose tissue in coronary arteries ranging from near normal to advanced atheromatous plaques. This allowed us to assess the immune cell phenotypes throughout disease progression in all specific layers of the arterial wall.

Our study yielded several significant findings, beginning with the identification of spatial contexts that influence the expression of key targets. We successfully performed both unsupervised and supervised cell typing across various disease stages, providing a comprehensive view of cellular and gene expression dynamics. Pathway analyses revealed that distinct cell types exhibit pathway enrichment based on their location, such as between the adventitia and plaque, as well as across different levels of disease severity. Additionally, unsupervised clustering of the datasets identified distinct transcriptomic signatures that change with disease severity and spatial context. UMAPs generated through CosMx™ revealed important differences in cell typing across severities, particularly the significant phenotypic variation in smooth muscle cell type 1 between early and advanced lesions. A new population of endothelial cells also emerged in advanced lesions, warranting further investigation. We have clearly demonstrated a significant expansion of adaptive immune cell subsets in advanced lesions. This finding is consistent with recent studies (Fernandez et al, 2019b) and reinforces the role of adaptive immunity in the pathology (Graybill et al, 1983). We also constructed cell-cell interaction networks and conducted neighbourhood analyses, which identified unique cellular clusters based on cell type composition and spatial coordinates. Nevertheless, more in-depth analyses of our combined GeoMx®/CosMx™ datasets, which we now make available to the community, are expected to further elucidate the precise roles of these immune cells and the specific pathways identified through our datasets and/or their integration with other publicly available data, in the pathogenesis of atherosclerosis across different stages of disease progression and vascular layers. This will enhance understanding of potential novel immunological targets, including the optimal timing and location for their therapeutic targeting or use in patient stratification, provided advancements in drug delivery platforms and diagnostic molecular imaging tools continue to progress.

Furthermore, we have attempted and believe we have a successful workflow for the integration of GeoMx® and CosMx™ technologies once a whole transcriptome atlas panel is released for CosMx™, which would enhance our overall understanding of the cellular and molecular landscape.

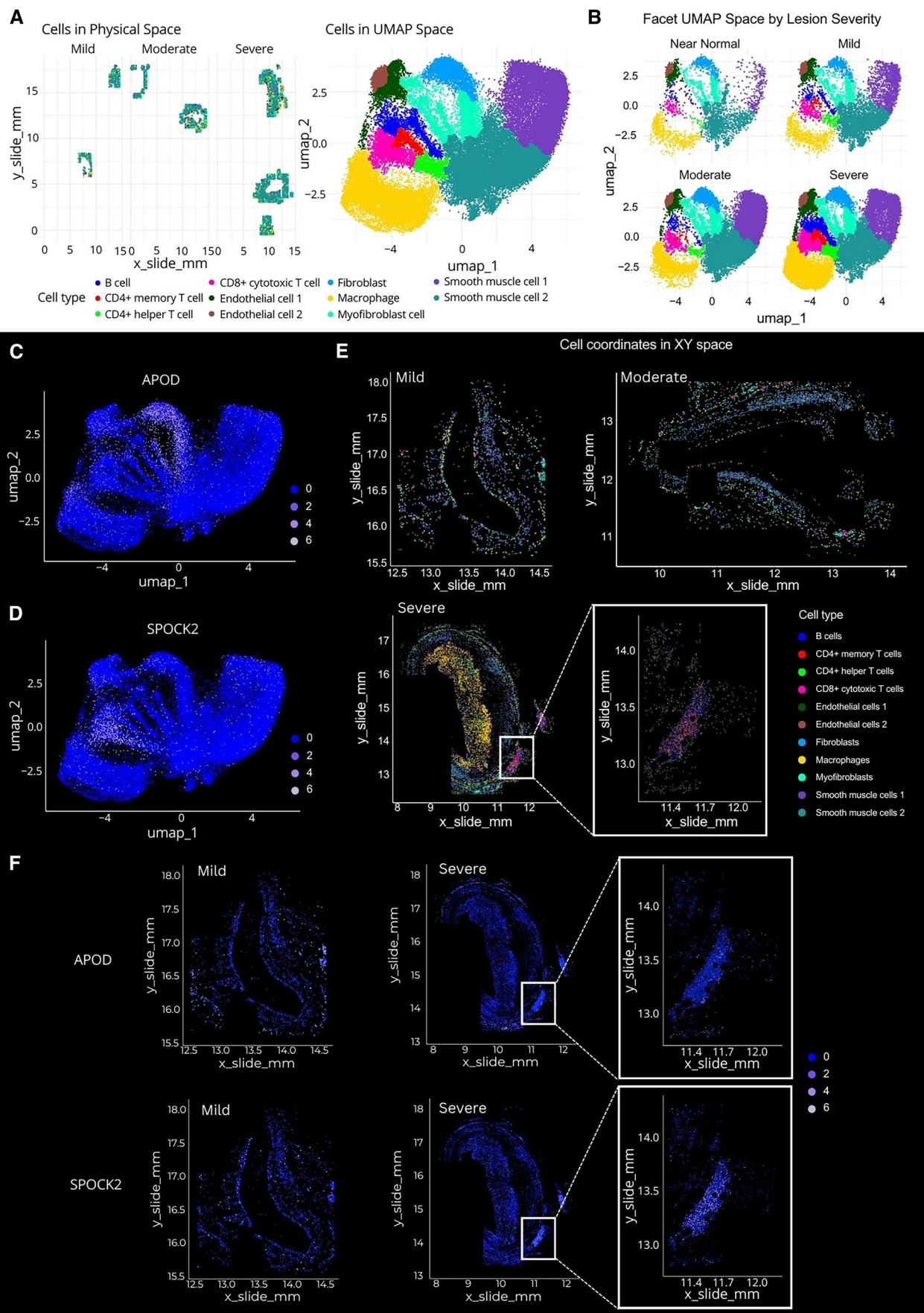

**Figure 5. Single-cell typing and target expression in CosMx datasets.**

(A) Single cells from the entire post-QC dataset are identified in both the slide physical and UMAP spaces by their cell type, following supervised cell typing using Census vasculature signature. (B) UMAP plots for the dataset split by lesion severity. (C, D) APOD (C) and SPOCK2 (D) expression in individual cells are represented in the entire dataset UMAP plots. (E) Representative mild, moderate and severe lesions showing single cell typing in situ. An ATLO region is highlighted at a higher magnification to reveal its cellular composition available through supervised cell typing. (F) Single cell APOD and SPOCK2 expression levels in representative mild and severe lesions (including an ATLO). Source data are available online for this figure.

We discussed the potential to improve cell type delineation by expanding the panel of targets. Although single-cell sequencing often results in a significant loss of targets, increasing the panel plexity to the whole transcriptome on the CosMx™ platform will not render this workflow obsolete. This is because GeoMx® sequencing provides deeper insights, allowing us to investigate rarer targets across different cell populations.

Recent advancements in integrating CosMx™ with single-cell RNA sequencing (scRNA-seq) have shown promising results (Chen et al, 2024), offering valuable insights for understanding complex biological processes. By exploring how these integration methods enhance our comprehension of atherosclerosis, we can better position our findings within the broader context of current research and methodologies.

At the time of this study, the largest commercially available CosMx™ panel was the Universal Cell Characterisation 1,000-plex. Currently, a 6,000-plex panel is available, and NanoString has publicly released data using the whole transcriptome panel for CosMx™, though it is expected to be commercially available at a later stage. As a result, future GeoMx® and CosMx™ integration will be feasible, providing the deepest cell typing and gene expression at the single-cell level available, once the whole transcriptome panel becomes commercially available from NanoString.

In conclusion, our study provides a comprehensive framework for understanding cell type dynamics and immune responses in human atherosclerosis at a spatially single-cell resolution. The insights gained from this spatial transcriptomics approach offer valuable information for identifying and understanding novel pathways involved in the pathogenesis of human atherosclerosis, particularly in specific anatomical vascular niches and throughout its development and associated cardiovascular complications.

## Methods

### Reagents and tools table

| Reagent/resource | Reference or source | Identifier or catalogue number |
|---|---|---|
| **Experimental models** | | |
| FFPE sections of human coronary arteries | University of Birmingham Human Biomaterial Resource Centre | RG_HBRCMV031 |
| **Recombinant DNA** | | |
| N/A | | |
| **Antibodies** | | |
| SYTO13 (GeoMx) | NanoString | Cat# 121300303 |
| Anti-CD45 (GeoMx) | NanoString | Cat# 121300310 |

| Reagent/resource | Reference or source | Identifier or catalogue number |
|---|---|---|
| Anti-CD4 (GeoMx) | Abcam | Cat# ab196147 |
| Anti-B2M/CD278 (CosMx) | NanoString | Cat# 121500020 |
| Anti-PanCK (CosMx) | NanoString | Cat# 121500021 |
| Anti-CD45 (CosMx) | NanoString | Cat# 121500021 |
| **Oligonucleotides and other sequence-based reagents** | | |
| Whole human-transcriptome atlas RNA panel for GeoMx | NanoString | Cat# 121401102 |
| CosMx universal cell characterisation RNA panel | NanoString | Cat# 121500002 |
| **Chemicals, enzymes and other reagents** | | |
| N/A | | |
| **Software** | | |
| GeoMx NGS pipeline software (version 2.0.0.16) | NanoString | |
| GeoMx® DSP (version 3.1.0.194) | NanoString | |
| SpatialOmicsOverlay | Bioconductor Bioconductor - SpatialOmicsOverlay Griswold, 2024 | |
| R (version 4.4.1) | CRAN | |
| CIBERSORT | Newman et al, 2015 | |
| AtoMx SIP (version 1.3) | NanoString | |
| Census | Github—sjdlabgroup GitHub—sjdlabgroup/Census: Accurate, deep, fast, and fully-automated scRNA-seq cell-type annotation Ghaddar and De, 2023 | |
| **Other** | | |

## Samples

This study utilised nine complete circumferential lengths of human coronary artery, including adventitia and surrounding adipose tissue, from three donors (one female; two male), obtained from hearts removed at the time of cardiac transplantation. Three samples from the same formalin-fixed coronary artery were embedded in each paraffin block, and serial sections were cut from

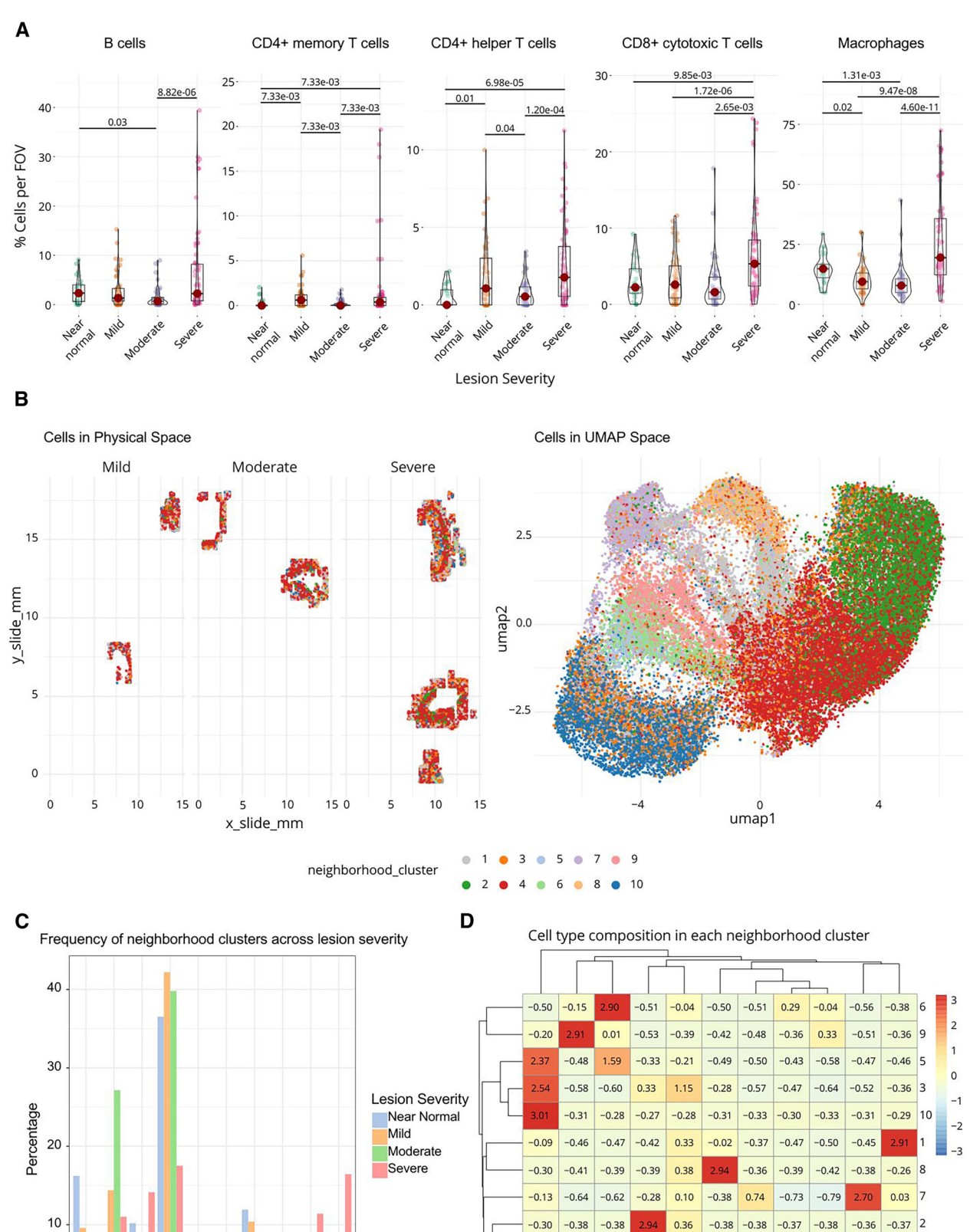

**Figure 6. Quantifying cellular microenvironments across lesion severity.**

(A) Violin plots depicting the percentage of different immune cell types (B cells, CD4+ memory T cells, CD4+ helper T cells, CD8+ cytotoxic T cells and macrophages) per field of view (FOV) across lesion severity (near normal, mild, moderate and severe). Only significant differences between cell densities across severities are shown (indicated by $P$ values and tested with a Kruskal–Wallis non-parametric $T$- test. Dunn's test was used for the pairwise test); $n = 25$ (near normal), 51 (mild), 52 (moderate) and 70 (severe). Box plots demonstrate the median as the centre line and the interquartile ranges (25th and 75th percentiles) as the enclosing box. Whiskers demonstrate the data range. (B) Cellular neighbourhood clusters were defined using the frequency of annotated populations within 20 mm of one another as input to clustering. Cellular neighbourhood clusters are shown in both physical space (left) and UMAP space (right). (C) Bar plot showing the frequency of neighbourhood clusters across lesion severity (near normal, mild, moderate and severe). Each bar represents the frequency of a cellular neighbourhood within a lesion severity. (D) Heatmap of cell type composition within each neighbourhood cluster. The heatmap illustrates the relative enrichment or depletion of different cell types within each cluster, with colours representing the z-score of enrichment (red – high, blue - low). Source data are available online for this figure.

each block. The coronary artery cross-sections varied within the same artery, and the three donors were chosen to provide a range from near normal intima to severe atherosclerotic plaques. No blinding or randomisation was performed in this study.

Ethics was obtained for method validation from the University of Birmingham Human Biomaterial Resource Centre (HBRC) RG_HBRCMV031. Informed consent was obtained from all subjects, and the experiments performed conformed to the principles set out in the WMA Declaration of Helsinki and the Department of Health and Human Services Belmont Report.

## GeoMx® digital spatial profiling (DSP)

For GeoMx®, 5 μm FFPE sections were stained with a cocktail of morphology markers (SYTO13 [NanoString, Cat# 121300303] at 1:10, CD45 [NanoString Cat# 121300310] at 1:40, and CD4 [Abcam Cat# ab196147] at 1:50) and hybridised with the whole human–transcriptome atlas RNA panel (WTA, 18.677 probes). A total of 55 regions of interest (ROIs) were placed across the nine cross-sections, covering regions that were CD45+CD4⁻ or CD45+CD4⁺ in the adventitia or the atherosclerotic plaque. As controls, a few ROIs were drawn on CD45−CD4− regions (smooth muscle or nerve). The collected barcodes were processed and used to prepare a cDNA library for Next Generation Sequencing (NGS). The NGS readout data was converted from FASTQ to DCC format using the GeoMx NGS pipeline (version 2.0.0.16), and the data was uploaded to the GeoMx® DSP.

The whole slide scans were generated by the GeoMx® DSP at the time of ROI placement and oligo collection and can be downloaded as OME. TIFF files were used for downstream analyses in packages such as the SpatialOmicsOverlay.

## GeoMx® data analysis

Data analysis was conducted on GeoMx® Digital Spatial Profiler (DSP) version 3.1.0.194 following the user manual recommended workflow and settings. Raw data from 55 ROIs profiled across three slides (each containing three artery sections as demonstrated in Fig. 1A) were subjected to technical and sequencing QC to assess the sequencing and overall quality of the data. Out of the 55 ROIs, 49 passed QC and were included in the downstream analysis. Of the six ROI segments that failed QC, two segments failed due to sequencing quality metrics, and the remainder failed due to low surface area and/or nuclei count. No segments were excluded at this step. Further, Probe QC was performed to identify negative control probe outliers, and none were reported. Filtering steps to exclude low-expressing targets and segments were conducted, which

eliminated 14,962 targets and two segments low in signal relative to the background. About 3715 targets and 53 segments remained for further data analysis. These data were normalised using Q3 normalisation and further subjected to unsupervised hierarchical clustering based on Pearson distance, differential gene expression, and pathway analysis. Pathway analysis was performed in R (version 4.4.1) using the fgsea (fast gene set enrichment analysis) R package (Korotkevich et al, 2021) and the Biological Processes section of the Gene Ontology database. GO pathways were accessed through R using the msigdbr R package (Dolgalev, 2021). Pathways were filtered based on a minimum and maximum size of 15 and 500, respectively, and required to have a minimum of 20% of their targets covered by the GeoMx® data. Pathways were visualised using GOCircle plots using the GOplot R package (Walter et al, 2015), and Cytoscape (Lotia et al, 2013), using the R package RCy3 (Gustavsen et al, 2019) and the Cytoscape plugin enrichment Map (Merico et al, 2010).

## Integration of GeoMx® dataset with the scRNA-seq dataset

To estimate the relative proportions of immune cell types within each ROI, we utilised CIBERSORT (Newman et al, 2015), integrating our GeoMx® dataset with previously published scRNA-seq data derived from coronary arteries of patients undergoing heart transplantation (GSE252243). CD45+ (PTPRC-expressing) clusters were pre-selected and annotated using feature genes provided by the original authors (Barcia Durán et al, 2024). Results were presented as the mean immune cell proportion per ROI within each group.

## CosMx™ spatial molecular imaging (SMI)

For CosMx™, serial 5 μm FFPE sections (to those used for GeoMx) were hybridised with the CosMx™ Universal Cell Characterisation RNA Panel containing 950 core targets; this panel was then customised with 20 additional targets to cover genes and pathways highlighted by the GeoMx® study. We placed 235 Fields of View (FOVs) across the nine cross-sections to profile the full thickness of the artery layers, atherosclerotic plaques and immune cell infiltrates. RNA readout of placed FOVs included an incubation with the reporter pool, followed by a reporter wash buffer to remove any unbound reporter probes. The imaging buffer was then released into the flow cell for imaging. Eight Z-stack images with a 0.8-μm step size for each FOV were taken. Photocleavable linkers on the fluorophores of the reporter probes were released by UV illumination and washed off with strip wash buffer. The fluidics and

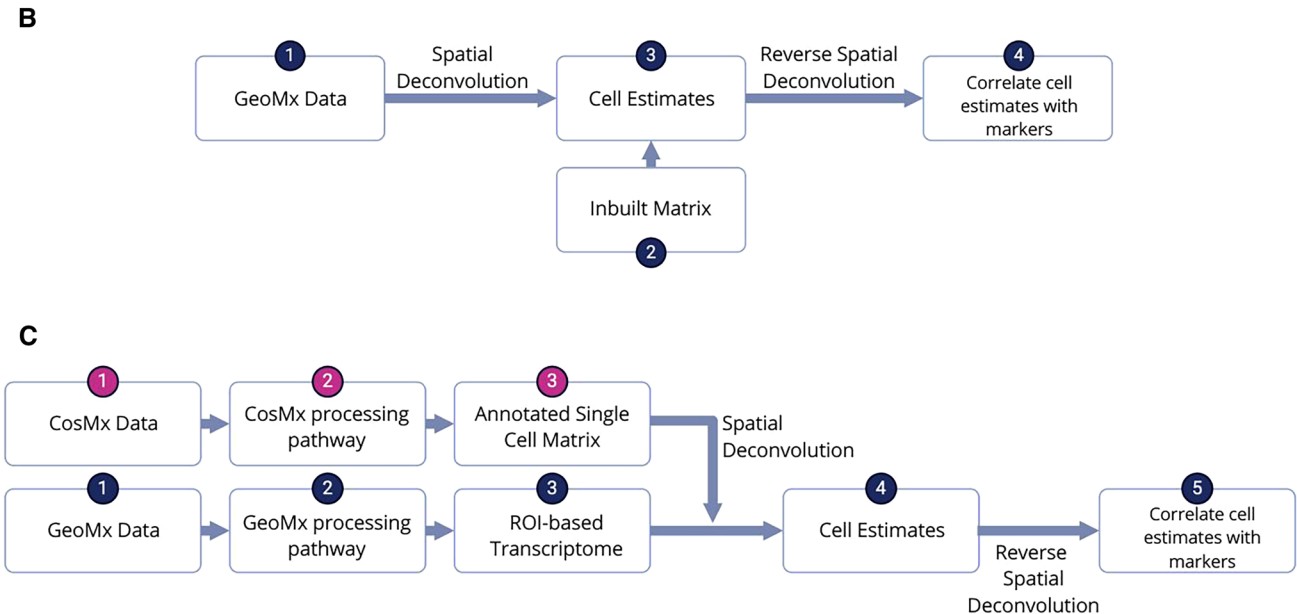

◀ **Figure 7. Integrating GeoMx® and CosMx™ datasets: the current and the future.**

(A) In situ expression of ANXA2 in a representative severe lesion in GeoMx® (top panel) and CosMx™ (bottom panel). Higher magnification boxes show ANXA2 expression in an ATLO. Figure 1C was reused to create this panel. (B) NanoString-provided workflow for spatial deconvolution of GeoMx® data. (C) Proposed workflow for spatial deconvolution utilising a CosMx-generated single cell matrix (from the same tissue) to inform cell estimates. Source data are available online for this figure.

imaging procedure was repeated for the 16 reporter pools, and the 16 rounds of reporter hybridisation-imaging were repeated multiple times to enhance RNA detection sensitivity.

## CosMx™ data analysis

Data analysis was initially performed using the AtoMx™ spatial informatics platform (SIP) version 1.2, following the user manual instructions. However, initial segmentation was qualitatively deemed not suitable for downstream processing, and cells were re-segmented with AtoMx™ SIP version 1.3. Post re-segmentation, cells and FOVs were subjected to Cell QC utilising the number of counts per cell (minimum 20), the proportion of negative probe counts (maximum 0.1), the count distribution (maximum −1) and the area outlier test (Grubbs test to flag cellular outliers based on cell area $[P < 0.01]$) to determine poor quality cells. FOV QC utilised the average counts per FOV to determine low-quality FOVs (minimum average counts – 35). 67% of cells and 85% of FOVs remained available for downstream processing.

Due to nuances with the AtoMx™ software, all further data exploration was performed outside of the AtoMx™ platform in the statistical programming language—R (version 4.4.1). A Seurat object was exported to Propath's AWS S3 bucket and downloaded for further exploration. Data were first filtered for cells and FOVs

passing quality control criteria and log-normalised using the Seurat R package (version 5.1.0) (Hao et al, 2024). Normalised data was scaled and subjected to linear (principal component analysis—PCA) and non-linear dimension reduction (uniform manifold approximation and projections—UMAP) for visualisation. Data were qualitatively checked for obvious batch effects and were harmonised to increase data integration for downstream clustering using Harmony (Korsunsky et al, 2019). A range of clustering mechanisms was evaluated, including Leiden (at a range of resolutions), InSituType (Danaher et al, 2022), and automatic cell typing using Census (Ghaddar and De, 2023). After evaluating each clustering mechanism by its tendency to fit the contours of the UMAP, Census was chosen as the clustering/annotation method for downstream analyses. Census is a pre-trained model using the Tabula Sapiens dataset, containing a "Vasculature" subset and allows for automatic cell typing of the data. Downstream analyses revolved around the Lesion Severity variable, providing outputs such as cell density comparisons, neighbourhood analysis (using the RANN R package (Jefferis, 2024)), single gene enrichment analysis, and multi-marker enrichment analysis.

## Data availability

The GeoMx® and CosMx™ datasets are available at https://www.ncbi.nlm.nih.gov/geo/query/acc.cgi?acc=GSE277170 and at https://www.ncbi.nlm.nih.gov/geo/query/acc.cgi?acc=GSE277441.

The source data of this paper are collected in the following database record: biostudies:S-SCDT-10_1038-S44321-025-00280-w.

## Peer review information

## The paper explained

### Problem
Atherosclerosis is a major cause of death due to the narrowing and stiffening of blood vessels. Innate and adaptive immunity are recruited to the vessel and are involved in the formation of the atherosclerotic plaque. However, the roles of immune cells during disease progression remain poorly understood. A definition of the spatial map of immune cells, and their transcriptome at the single-cell level, in the human plaque, could aid in the understanding of the immune response's contribution to atherosclerosis progression.

### Results
GeoMx® and CosMx™ are powerful technologies to study tissue transcriptomics. We used them to analyse serial sections of human coronary arteries from patients with varying degrees of atherosclerotic lesion severity, with a focus on immune cells and pathways. We generated pathway analyses, cell typing and neighbourhood analyses, as well as integrating findings from both datasets, to identify biological processes all the way to the single-cell level.

### Impact
Our approach unbiasedly identifies molecules and pathways of relevance to the understanding of atherosclerosis pathogenesis and could support studies in the near future that will lead to the identification of novel therapeutic targets to better control the deadly outcomes of atherosclerosis.

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

## Acknowledgements

We thank HBRC at the University of Birmingham for providing the human tissues. We thank Nick Jones at the University of Swansea for the critical reading of the manuscript. This study was funded by C.Mauro's British Heart Foundation Senior Basic Science Research Fellowship FS/SBSRF/22/31031 and by Propath UK. C.Mauro is supported by UNION-HORIZON-MSCA-DN-2024-111167421. PM is supported by British Heart Foundation grants (PG/19/84/34771, FS/19/56/34893 A, PG/21/10541, PG/21/10634 and PG/24/11946), the Italian Ministry of University and Research (MIUR) PRIN 2022 (2022T45AXH) funded by the European Union—Next Generation EU, Mission 4, Component 1, CUP E53D23012760006, and the European Union—Next Generation EU, Project CN00000041, Mission 4, Component 2, CUP B93D21010860004. This independent research was carried out at Propath UK Ltd and the National Institute for Health and Care Research (NIHR) Birmingham Biomedical Research Centre (BRC). The views expressed are those of the authors and not necessarily those of the above-listed funders.

## Author contributions

**Joana Campos**: Conceptualisation; Resources; Data curation; Software; Formal analysis; Supervision; Validation; Investigation; Visualisation; Methodology; Writing—original draft; Writing—review and editing. **Jack L McMurray**: Resources; Data curation; Software; Formal analysis; Validation; Investigation; Visualisation; Methodology; Writing—review and editing. **Michelangelo Certo**: Investigation; Visualisation; Writing—review and editing. **Ketaki Hardikar**: Investigation; Visualisation; Writing—review and editing. **Chris Morse**: Investigation; Writing—review and editing. **Clare Corfield**: Investigation; Writing—review and editing. **Bettina M Weigand**: Investigation; Writing—review and editing. **Kun Yang**: Resources; Software; Investigation; Methodology; Writing—review and editing. **Mohsen Shoaran**: Resources; Software; Investigation; Methodology; Writing—review and editing. **Thomas D Otto**: Resources; Software; Investigation; Visualisation; Methodology; Writing—review and editing. **Desley Neil**: Conceptualisation; Investigation; Methodology; Writing—review and editing. **Pasquale Maffia**: Conceptualisation; Resources; Data curation; Software; Supervision; Investigation; Visualisation; Writing—review and editing. **Claudio Mauro**: Conceptualisation; Data curation; Supervision; Funding acquisition; Writing—original draft; Writing—review and editing.

Source data underlying the figure panels in this paper may have individual authorship assigned. Where available, figure panel/source data authorship is listed in the following database record: biostudies:S-SCDT-10_1038-S44321-025-00280-w.

## Disclosure and competing interests statement

The authors declare no competing interests.

# Expanded View Figures

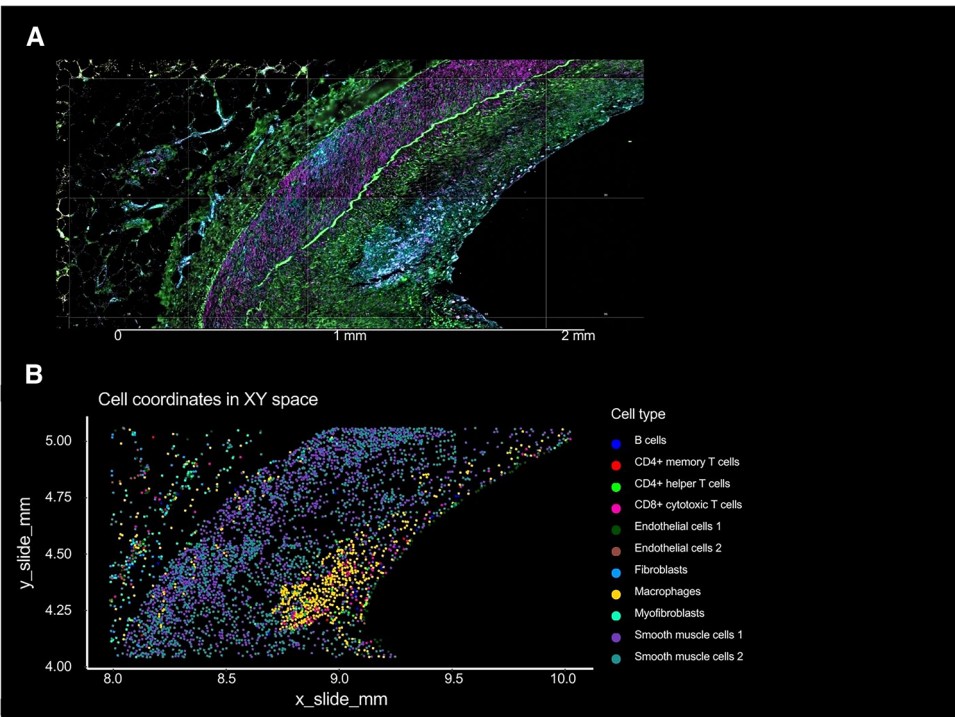

**Figure EV1.   Protein immunofluorescence staining and RNA-driven cell typing of the same region within an atherosclerotic vessel.**

(A) Immunofluorescence scan generated by CosMx™ showing the full thickness of the coronary artery and (B) respective cell typing equivalent following supervised cell typing using Census vasculature signature.

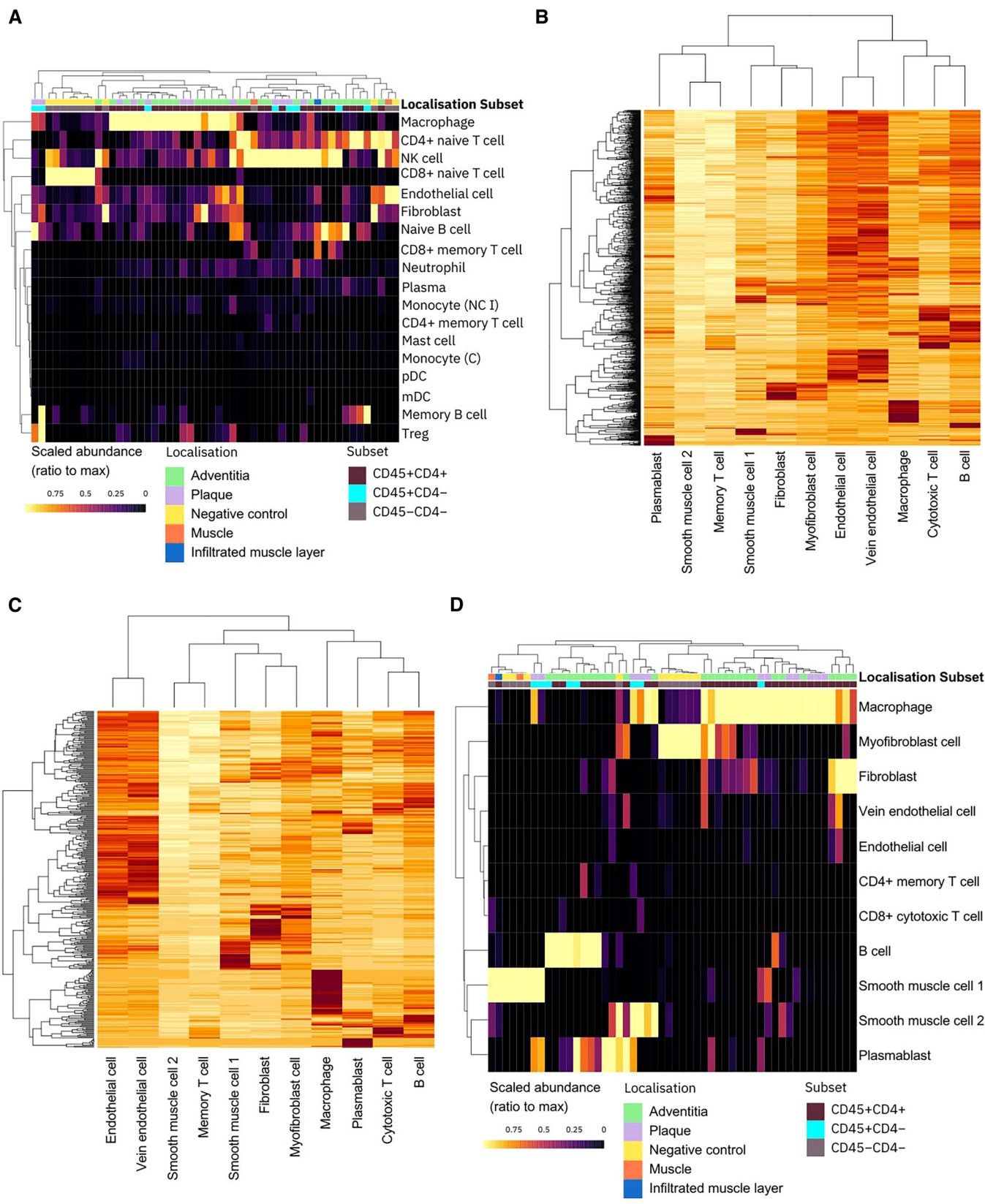

**Figure EV2. A comparison of spatial deconvolution estimates using inbuilt matrices versus CosMx™ derived signatures.**

(A) Heatmap of spatial deconvolution estimates using the inbuilt GeoMx® reference matrix 'safeTME'. Scaled abundances are shown as a ratio to the maximum value are displayed across five tissue localisations (adventitia, plaque, negative control, muscle and infiltrated muscle layer). 'Subsets' correspond to ROIs segmented on the GeoMx® platform and are highlighted. Cell types present in the safeTME matrix are indicated as rows. (B) Heatmap of the CosMx™-derived cell signature matrix. Census-annotated cell populations from the CosMx™ are represented as columns with rows representing genes on the CosMx™ platform. Genes are scaled from red to white, with red indicating a higher expression. (C) Heatmap of the genes present in the GeoMx® dataset from the CosMx™-derived cell signature matrix. (D) Heatmap of spatial deconvolution estimates using the CosMx™-derived matrix. Cell types present in the CosMx™-matrix are indicated as rows.

