## [Peer Review File · EMBO Molecular Medicine]

Spatial transcriptomics elucidates localized immune responses in atherosclerotic coronary artery

Joana Campos, Jack McMurray, Michelangelo Certo, Ketaki Hardikar, Chris Morse, Clare Corfield, Melanie Weigand, Kun Yang, Mohsen Shoaran, Thomas Otto, Desley Neil, Pasquale Maffia, and Claudio Mauro

Corresponding authors: Claudio Mauro (c.mauro@bham.ac.uk) , Joana Campos (joana.campos@propath.co.uk)

Review Timeline:

Submission Date:	29th Nov 24
Editorial Decision:	11th Feb 25
Revision Received:	23rd May 25
Editorial Decision:	20th Jun 25
Revision Received:	8th Jul 25
Accepted:	11th Jul 25

Editor: Lise Roth

Transaction Report:

11th Feb 2025

Dear Claudio,

Thank you for the submission of your manuscript to EMBO Molecular Medicine, and please accept my renewed apologies for the exceptional delay in getting back to you. As explained in a previous correspondence, we encountered difficulties obtaining reports from the referees and eventually had to secure an additional referee. We have now heard back from the 3 referees who reviewed your manuscript.

As you will see from the reports below, referees #1 and #2 acknowledge the technological advancement and quality of the study, but also criticize the lack of biological insight. Referee #4 finds the dataset useful, nevertheless also requests additional analyses to further demonstrate the significance of the data.

Following further consultation with the referees, we clarified that the manuscript was intended as a Resource, with the understanding that the new biological insight could be limited. The referees agreed that your work would potentially constitute a useful resource for the community, and we will therefore welcome the submission of your revised manuscript according to the referees' recommendations.

Addressing the referees' concerns in full will be necessary for further considering the manuscript in our journal, and acceptance of the manuscript will entail a second round of review. EMBO Molecular Medicine encourages a single round of revision only and therefore, acceptance or rejection of the manuscript will depend on the completeness of your responses included in the next, final version of the manuscript. For this reason, and to save you from any frustrations in the end, I would strongly advise against returning an incomplete revision.

We are expecting your revised manuscript within three months, if you anticipate any delay, please contact us.

We require:

- 1) A .docx formatted version of the manuscript text (including legends for main figures, EV figures and tables). Please make sure that the changes are highlighted to be clearly visible.
- 2) Individual production quality figure files as .eps, .tif, .jpg (one file per figure). For guidance, download the 'Figure Guide PDF' (<https://www.embopress.org/page/journal/17574684/authorguide#figureformat>).
- 3) At EMBO Press we ask authors to provide source data for the main figures. Our source data coordinator will contact you to discuss which figure panels we would need source data for and will also provide you with helpful tips on how to upload and organize the files.
- 4) A .docx formatted letter INCLUDING the reviewers' reports and your detailed point-by-point responses to their comments. As part of the EMBO Press transparent editorial process, the point-by-point response is part of the Review Process File (RPF), which will be published alongside your paper.
- 5) A complete author checklist, which you can download from our author guidelines (<https://www.embopress.org/page/journal/17574684/authorguide#submissionofrevisions>). Please insert information in the checklist that is also reflected in the manuscript. The completed author checklist will also be part of the RPF.
- 6) All Materials and Methods need to be described in the main text using our 'Structured Methods' format. According to this format, the Methods section includes a Reagents and Tools Table (listing key reagents, experimental models, software and relevant equipment and including their sources and relevant identifiers) followed by a Methods and Protocols section describing the methods, ideally using a step-by-step protocol format. The aim is to facilitate adoption of the methodologies across labs. Please download and fill our Reagents and Tools Table template (.docx), which you can find in our author guidelines: <https://www.embopress.org/page/journal/14693178/authorguide#structuredmethods>. When submitting your revised manuscript, please do not include the Reagents and Tools Table in the Methods section of the manuscript but upload it as a separate file choosing the file type "Reagent Table".

An example of a Method paper with Structured Methods can be found here:
<https://www.embopress.org/doi/10.15252/msb.20178071>

7) Please note that all corresponding authors are required to supply an ORCID ID for their name upon submission of a revised manuscript.

8) It is mandatory to include a 'Data Availability' section after the Materials and Methods. Before submitting your revision, primary datasets produced in this study need to be deposited in an appropriate public database, and the accession numbers and database listed under 'Data Availability'. Please remember to provide a reviewer password if the datasets are not yet public (see <https://www.embopress.org/page/journal/17574684/authorguide#dataavailability>).

9) For data quantification: please specify the name of the statistical test used to generate error bars and P values, the number (n) of independent experiments (specify technical or biological replicates) underlying each data point and the test used to calculate p-values in each figure legend. The figure legends should contain a basic description of n, P and the test applied. Graphs must include a description of the bars and the error bars (s.d., s.e.m.). Please provide exact p values.

10) Our journal encourages inclusion of *data citations in the reference list* to directly cite datasets that were re-used and obtained from public databases. Data citations in the article text are distinct from normal bibliographical citations and should directly link to the database records from which the data can be accessed. In the main text, data citations are formatted as follows: "Data ref: Smith et al, 2001" or "Data ref: NCBI Sequence Read Archive PRJNA342805, 2017". In the Reference list, data citations must be labeled with "[DATASET]". A data reference must provide the database name, accession number/identifiers and a resolvable link to the landing page from which the data can be accessed at the end of the reference. Further instructions are available at .

11) We replaced Supplementary Information with Expanded View (EV) Figures and Tables that are collapsible/expandable online. A maximum of 5 EV Figures can be typeset. EV Figures should be cited as 'Figure EV1, Figure EV2' etc... in the text and their respective legends should be included in the main text after the legends of regular figures.

12) The paper explained: EMBO Molecular Medicine articles are accompanied by a summary of the articles to emphasize the major findings in the paper and their medical implications for the non-specialist reader. Please provide a draft summary of your article highlighting

13) Author contributions: CRediT has replaced the traditional author contributions section because it offers a systematic machine readable author contributions format that allows for more effective research assessment. Please remove the Authors Contributions from the manuscript and use the free text boxes beneath each contributing author's name in our system to add specific details on the author's contribution. More information is available in our guide to authors.

Please also suggest a visual abstract to illustrate your article as a PNG file 550 px wide x 300-600 px high. A cropped portion of this image will serve as thumbnail for the table of content on our webpage.

16) As part of the EMBO Publications transparent editorial process initiative (see our Editorial at <http://embomolmed.embopress.org/content/2/9/329>), EMBO Molecular Medicine will publish online a Review Process File (RPF) to accompany accepted manuscripts.

In the event of acceptance, this file will be published in conjunction with your paper and will include the anonymous referee reports, your point-by-point response and all pertinent correspondence relating to the manuscript. Let us know whether you agree with the publication of the RPF and as here, if you want to remove or not any figures from it prior to publication. Please note that the Authors checklist will be published at the end of the RPF.

I look forward to receiving your revised manuscript.

Yours sincerely,

Lise Roth

***** Reviewer's comments *****

Referee #1 (Remarks for Author):

This study introduces a novel application of the combined NanoString GeoMx® and CosMx{trade mark, serif} platforms to spatially resolve cell types and gene expression across different stages of atherosclerosis in human coronary arteries, providing critical insights into the localized immune mechanisms involved at each stage of the disease. A major strength is the technological advancement, but a limitation is that it remains unclear for what is it good? What did we learn about human atherosclerosis? In the end, while the study is interesting, the work remains descriptive, and the medical or therapeutic relevance remains unclear. The following are comments and suggestions for improvement:

Minor comments:

1. What specific strategies and criteria were employed in selecting and setting the Regions of Interest (ROIs) in this study? Please provide a more detailed explanation.
2. In the section Single cell spatial transcriptomic resolution using CosMx Spatial Molecular Imager, how was quality control (QC) conducted? Please elaborate on the QC process.
3. While the authors state that data were collected for the full thickness of the vessel and the vasa vasorum, why does the analysis in Figures 2 and 3 focus only on comparing the adventitia and atherosclerotic plaques? Additionally, in the subsequent analysis of early and advanced lesions, the authors emphasize immune cells in the adventitia but do not provide insights into other layers or components. Please clarify this limitation.
4. In the "Single cell typing provided by Census" section, the study identifies cell type enrichment and transcriptomic similarities among different cell types. While the role of specific immune cells in advanced lesions is well illustrated, their functional significance within the plaque microenvironment, especially in early lesions, remains unclear. Please discuss their roles in relation to clinical or pathological significance.
5. Please ensure that all microscopy images include scale bars or magnification details.
6. It is recommended that the discussion on the clinical implications of the study findings be strengthened, particularly how the immune cell dynamics identified using the combined platforms could inform therapeutic strategies for addressing atherosclerosis progression.

Referee #2 (Comments on Novelty/Model System for Author):

This is a well performed study, using Geomx and cosm to unravel the location and subsets of immune cells in early and advanced stages of coronary atherosclerosis. The authors include intima, media, adventitia and peripheral adipose tissue. The data analysis is excellent. However the biological information obtained from this study is purely descriptive and limited.

Referee #2 (Remarks for Author):

This is a well performed study, using Geomx and cosm to unravel the location and subsets of immune cells in early and advanced stages of coronary atherosclerosis. The authors include intima, media, adventitia and peripheral adipose tissue. The data analysis is excellent. However the biological information obtained from this study is largely descriptive.

Comments:

- how do the immune cell subsets observed in the goemx/cosmx spatial analysis correlate with those that have been observed in snRNAseq and scRNAseq in coronary artery plaques and carotid endarterectomy?
- a slide is only 5 um, how do the data vary in:
 - * adjacent sections for the same technique
 - * the 3 different sites within the same coronary artery
 - * what is the inter-patient variability
- The data are very descriptive. Biological validation of the major pathways that have been discovered is missing.

Referee #4 (Remarks for Author):

Campos et al. provided a valuable resource by generating CosMx and GeoMx data across different stages of atherosclerosis. They conducted differential gene expression analysis, gene set enrichment analysis, and neighborhood analysis. While this work offers a useful dataset, additional analyses would further demonstrate the significance and usefulness of the generated data. Below are specific comments for consideration.

1. Line 104. Immune cell segments. This can mislead. The segments defined in the manuscript can contain many cell types. This has to be explained when explaining the results.
2. In this regard, Fig 2b can mislead as well as segments may contain other cells and cell type portion (rather than immune cell gene expression) can determine DEGs. This has to be clearly explained. Otherwise, please show. Integration with public scRNAseq or CosMx data could explain it.
3. Similarly, it does not clearly tell that the source of change of APOD and SPOCK2 when explaining Fig 2. It is discussed later in page 5 (and Fig 5) more in detail. However, current description can be read that the changes are from CD45+CD4- or CD45+CD4+ population. Clear description that the GeoMx detects transcriptome from multiple cell types is required.
4. Fig 2b explains the genes differentially observed. Integration of scRNAseq (PMID:36224302 or any other scRNAseq data relevant for this study) can explain Fig 2b about the source cell type for this change.
5. Line 161-163: The unsupervised clustering does not have any meaning and can conflict with Fig 5e. Because Fig 5e is prepared with annotation, the UMAP plot in Fig 4e is not needed. Please remove.
6. Line 173-174. It is not always correct to use the distance in the UMAP plot to discuss transcriptomic similarity (<https://pair-code.github.io/understanding-umap/>). The distance between the sub-cell type can be calculated to show it. Otherwise, change the wording.
7. Neighborhood analysis if of potential interest. Are there any genes whose expression is influenced by cell-cell interactions? Do immune cells exhibit neighbor-dependent gene expression? CellChat, CellNeighborEx or any other tools can explain.
8. Deconvolution can be done by including public scRNAseq data. Even though the tissue are not perfectly matching, this approach does not suffer "limited overlap".

We thank the reviewers and the editor for an overall positive assessment of our resource paper and for the constructive suggestions for improvement they have provided. Below we address the specific points raised by each reviewer. All edits we made in response to editor's and reviewers' points are highlighted in yellow in the revised manuscript.

Referee #1 (Remarks for Author):

This study introduces a novel application of the combined NanoString GeoMx and CosMx platforms to spatially resolve cell types and gene expression across different stages of atherosclerosis in human coronary arteries, providing critical insights into the localized immune mechanisms involved at each stage of the disease. A major strength is the technological advancement, but a limitation is that it remains unclear for what is it good? What did we learn about human atherosclerosis? In the end, while the study is interesting, the work remains descriptive, and the medical or therapeutic relevance remains unclear. The following are comments and suggestions for improvement.

We thank the reviewer for the positive comment about our technological innovation applied to human atherosclerosis. In terms of novelty, we would like to emphasise that studies investigating the different stages of atherosclerosis progression in humans are generally lacking, whereby most human studies have focussed on end stage plaques from endarterectomies, where the vessel wall is generally lost. Here, in GeoMx we have focussed the analysis on CD45+ and CD4+ both in adventitia and core of the plaque whereas our CosMx analysis provides gene expression across the whole vessel layers. Our study is meant as a high-quality resource paper and not meant to validate any specific pathway. Nonetheless, we do believe this will become an important study for others in the atherosclerosis field to validate their pathways of interest, to spur integrations with other datasets, and also to inspire scientists in other fields to conduct similar combined approaches.

Minor comments:

1. What specific strategies and criteria were employed in selecting and setting the Regions of Interest (ROIs) in this study? Please provide a more detailed explanation.

In our GeoMx study, ROIs were placed with the aim of analysing both CD45+CD4- and CD45+CD4+ populations in the adventitia and core of the plaque across the various stages of atherosclerosis present in our cross-sections. To achieve this, we have stained the samples with anti-CD45 and anti-CD4 antibodies as morphology markers and used their intensity thresholds to segment ROIs into CD45+CD4- and CD45+CD4+ rich segments. As controls, we also collected few CD45-CD4- ROIs in nerve and muscle areas. All ROIs were placed under the guidance of Prof Neil, the pathologist who collected the tissues in the first instance and a senior author on the manuscript.

These aspects were clarified in the text at **lines 103-112**.

2. In the section Single cell spatial transcriptomic resolution using CosMx Spatial Molecular Imager, how was quality control (QC) conducted? Please elaborate on the QC process.

We have expanded the QC section for the CosMx SMI experiment and included the following text in the manuscript (**lines 454-459**): "Post re-segmentation, cells and FOVs were subjected to Cell QC utilising the number of counts per cell (minimum – 20), the proportion of negative probe counts (maximum 0.1), the count distribution (maximum -1) and the area outlier test (Grubbs test to flag cellular outliers based on cell area [P < 0.01]) to determine

poor quality cells. FOV QC utilised the average counts per FOV to determine low quality FOVs (minimum average counts – 35). 67% of cells and 85% of FOVs remained available for downstream processing.”

3. *While the authors state that data were collected for the full thickness of the vessel and the vasa vasorum, why does the analysis in Figures 2 and 3 focus only on comparing the adventitia and atherosclerotic plaques? Additionally, in the subsequent analysis of early and advanced lesions, the authors emphasize immune cells in the adventitia but do not provide insights into other layers or components. Please clarify this limitation.*

Since our primary interest in the study was on immune cells, most ROIs were collected in adventitia and core of the plaque which are the typical locations of immune cell accumulation. Subsequent analyses were focused on adventitia as a result of the striking differences in terms of immune cell signals when comparing early versus late disease. These aspects were already discussed but are now further clarified in the text at **lines 103-112**.

4. *In the "Single cell typing provided by Census" section, the study identifies cell type enrichment and transcriptomic similarities among different cell types. While the role of specific immune cells in advanced lesions is well illustrated, their functional significance within the plaque microenvironment, especially in early lesions, remains unclear. Please discuss their roles in relation to clinical or pathological significance.*

The reviewer is correct, and we have added details at **lines 199-204**.

5. *Please ensure that all microscopy images include scale bars or magnification details.*

We have revised accordingly.

6. *It is recommended that the discussion on the clinical implications of the study findings be strengthened, particularly how the immune cell dynamics identified using the combined platforms could inform therapeutic strategies for addressing atherosclerosis progression.*

We had already touched upon how our study could inform therapeutic strategies at **lines 340-348**, which we now have further expanded.

Referee #2 (Remarks for Author):

This is a well performed study, using Geomx and cosm to unravel the location and subsets of immune cells in early and advanced stages of coronary atherosclerosis. The authors include intima, media, adventitia and peripheral adipose tissue. The data analysis is excellent. However the biological information obtained from this study is largely descriptive.

Our study is meant as a high-quality resource paper which is not meant to validate any specific pathway but we believe will be important for others in the atherosclerosis field to validate their pathways of interest, to spur integrations with other datasets but also to inspire scientists in other fields to conduct similar combined approaches.

Comments:

- how do the immune cell subsets observed in the goemx/cosmx spatial analysis correlate with those that have been observed in snRNAseq and scRNAseq in coronary artery plaques and carotid endarterectomy?

To address the reviewer's remark, we performed deconvolution analysis (estimating the proportions of different immune cell types in each group or condition) utilising publicly available single-cell RNA sequencing reference data (GSE252243; Barcia Duran et al., Nat Cardio Res 2024) from coronary arteries. As shown in the new panels **f & g** of **Figure 2** (results described at **lines 132-140**; methods described at **lines 428-434**), we observe an increase in T and B cells and a decrease in macrophages within the adventitia compared to the plaque. Similarly, we find a higher percentage of lymphocytes in the adventitia of severe or advanced lesions compared to normal arteries or those with mild atherosclerosis.

- a slide is only 5 um, how do the data vary in:

* adjacent sections for the same technique

We do acknowledge that adjacent sections of the same tissue will never be a perfect match; however, the structural differences from one section to the next one are minimal and hardly identifiable. Therefore, we are confident that the biological events taking place will be very similar across the sections used for the 2 techniques used in this study. ProPath does have experience from other studies of analysing serial sections of the same tissue with the same assay, which shows that no significant differences are observed. Furthermore, we have included H&E-stained serial sections to the ones used in the GeoMx WTA assay (Fig.1a) and from a morphological point of view these are very similar.

* the 3 different sites within the same coronary artery

The disease stage at the 3 different cross-sections from the same patient varies; some areas have no visible atherosclerosis, others present varying stages/grades of disease. We annotated each cross-section based on a pathologist's assessment and these annotations were used at the data analysis stage (this information was already included in the manuscript at **lines 103-112**).

* what is the inter-patient variability

Unfortunately, we have no access to patients' information via the Ethics protocol that covered this study.

- The data are very descriptive. Biological validation of the major pathways that have been discovered is missing.

As mentioned, and agreed with the editor, we do not mean to validate any specific pathway in this paper but to provide a resource that we and others can consult and validate pathways in follow up studies. We believe the datasets generated with this study will be important for others in the atherosclerosis field to validate their pathways of interest, to spur integrations with other datasets but also to inspire scientists in other fields to conduct similar combined approaches.

Referee #4 (Remarks for Author):

Campos et al. provided a valuable resource by generating CosMx and GeoMx data across different stages of atherosclerosis. They conducted differential gene expression analysis, gene set enrichment analysis, and neighborhood analysis. While this work offers a useful dataset, additional analyses would further demonstrate the significance and usefulness of the generated data.

We thank the reviewer for the overall positive comments to our study.

Below are specific comments for consideration.

1. Line 104. Immune cell segments. This can mislead. The segments defined in the manuscript can contain many cell types. This has to be explained when explaining the results.

We thank the reviewer for this comment, which highlighted the need to clarify the wording in the manuscript regarding the segments analysed. Wording to clarify that the samples areas profiled with the GeoMx assay were segmented based on their expression of CD45 and CD4 has been added to the revised text (**lines 103-112**). Therefore, whilst these would not be pure, segments were enriched for immune cells.

2. In this regard, Fig 2b can mislead as well as segments may contain other cells and cell type portion (rather than immune cell gene expression) can determine DEGs. This has to be clearly explained. Otherwise, please show. Integration with public scRNAseq or CosMx data could explain it.

As mentioned above, segments were enriched for CD45+ cells, and therefore differential gene expression observed in Fig 2b can be attributed, at least in part, to the immune cell compartment present in those segments.

We have addressed the suggested integration strategy at point 4 below.

3. Similarly, it does not clearly tell that the source of change of APOD and SPOCK2 when explaining Fig 2 . It is discussed later in page 5 (and Fig 5) more in detail. However, current description can be read that the changes are from CD45+CD4- or CD45+CD4+ population. Clear description that the GeoMx detects transcriptome from multiple cell types is required.

Volcano plot in Fig 2c shows that CD45+ segments from the adventitia exhibit significant different levels of APOD and SPOCK2 between early and advanced lesions. Further analysis revealed that the significant changes in APOD and SPOCK2 expression are attributable to the CD4+ compartment within the CD45+-enriched segments.

4. Fig 2b explains the genes differentially observed. Integration of scRNAseq (PMID:36224302 or any other scRNAseq data relevant for this study) can explain Fig 2b about the source cell type for this change.

To address the reviewer's remark regarding the source cell types underlying the observed differential gene expression between different layers of the vessel (Fig. 2b) and across varying levels of disease severity (Fig. 2c) in the GeoMx dataset, we performed deconvolution analysis (estimating the proportions of different immune cell types in each group or condition) utilising publicly available single-cell RNA sequencing reference data. Given that the dataset recommended by the reviewer (GSE159677; Alsaigh et al., Nat Methods 2015) is derived from human carotid arteries, we also applied the same workflow to a more closely matched tissue type (i.e., human coronary arteries [GSE252243; Barcia Duran et al., Nat Cardio Res 2024]) for comparison. As shown in **New Fig 2f-g** (coronary arteries; results described at **lines 132-140**; methods described at **lines 428-434**) and below (carotid arteries), the trends in immune cell composition changes are broadly similar across reference datasets despite differences in arterial tissue origin. We proposed incorporating only the coronary artery analysis into the manuscript, while the carotid artery analysis is here for the referee's reference.

Carotid artery analysis (GSE159677): a) Bar charts showing the estimated relative proportions of immune cell types in CD45-enriched (CD45+) segments located in the adventitia and in the plaque; **b)** Bar charts showing the estimated relative proportions of immune cell types in CD45-enriched (CD45+) segments located in the adventitia of early (near normal/mild) and advanced (moderate/severe) lesions.

5. Line 161-163: *The unsupervised clustering does not have any meaning and can conflict with Fig 5e. Because Fig 5e is prepared with annotation, the UMAP plot in Fig 4e is not needed. Please remove.*

We agree with the reviewer and have removed the panel and the related text.

6. Line 173-174. *It is not always correct to use the distance in the UMAP plot to discuss transcriptomic similarity (<https://pair-code.github.io/understanding-umap/>). The distance between the sub-cell type can be calculated to show it. Otherwise, change the wording.*

We agree with the reviewer and whilst global clusters can be used to comment on transcriptomic similarity, this is not always the case. **Lines 190-191** have been reworded to address this point.

7. *Neighborhood analysis if of potential interest. Are there any genes whose expression is influenced by cell-cell interactions? Do immune cells exhibit neighbor-dependent gene expression? CellChat, CellNeighborEx or any other tools can explain.*

We agree with the reviewer that neighborhood analysis would be an interesting avenue to explore further, however, it is outside of the remit of this project.

8. *Deconvolution can be done by including public scRNAseq data. Even though the tissue are not perfectly matching, this approach does not suffer "limited overlap".*

We have addressed this issue at point 4 above.

20th Jun 2025

Dear Prof. Mauro,

Thank you for submitting your revised study, which has been reviewed by the three initial referees. As you will see below, they are overall satisfied with the revisions, and I will therefore be able to accept your manuscript once the following editorial concerns are addressed:

1/ Referees' concerns:

Please address referee #4's remaining concern in writing.

2/ Manuscript text:

- Please remove the yellow highlights and only keep in track changes mode any new modification in the text.
- Please clarify whether Joanna Campos is a corresponding author as well and/or add the email address of Pasquale Maffia if he is a co-corresponding author on the manuscript title page.
- Please note that all corresponding authors are required to supply an ORCID ID for their name upon submission of a revised manuscript. An ORCID ID is currently missing for Campos (if she is a co-corresponding author).
- Please provide up to 5 keywords.
- Methods:
 - o Antibodies: please provide dilutions/concentrations
 - o Statistics: please provide a statement on blinding and randomization.
- Data Availability section: Please note that this section is restricted to new primary data that are part of this study.
- Funding should be merged with Acknowledgements. The complete list of funders should be both in the manuscript and the submission system.

3/ Figures:

- Figure re-use should be indicated in the figure legends (i.e. 1C vs. 2E, 1C vs. 7A).
- There is a reference to Suppl Table 1 in the text, please correct; the individual panels should be called out for Fig EV1.
- Table 1: please remove the legend from the manuscript text and add it to the excel file. Rename the table Dataset EV1.
- Please address the queries from our copy editors in the figure legends:
 1. Please indicate the statistical test used for data analysis in the legends of figures 2B, C
 2. Please note that the box plots need to be defined in terms of minima, maxima, centre, bounds of box and whiskers, and percentile in the legends of figures 2D, 6A
 3. Please note that information related to n is missing in the legends of figures 2B, C, D; 6A

4/ Thank you for providing Source Data. The files should be ZIPed and uploaded as one file per figure. If too large, we recommend uploading everything to BioStudies.

5/ Checklist:

Please fill in the entire section "Experimental study design and statistics"

6/ Please provide The paper explained, which is different from the abstract. It should be structured as follows: Medical Issue/Results/Clinical impact. Please refer to any of our published article for examples.

7/ Thank you for providing a synopsis text and image, however the image should be a schematic/visual abstract that illustrates the article, rather than a panel from your manuscript figure.

8/ As part of the EMBO Publications transparent editorial process initiative (see our Editorial at <http://embomolmed.embopress.org/content/2/9/329>), EMBO Molecular Medicine will publish online a Review Process File (RPF) to accompany accepted manuscripts.

We note that you agree with the publication of the RPF.

I look forward to receiving your revised manuscript.

Yours sincerely,

Lise Roth

Lise Roth, PhD

***** Reviewer's comments *****

Referee #1 (Comments on Novelty/Model System for Author):

The paper uses the latest spatial transcriptomics but neither have we learned much new nor is there immediate medical impact. I believe as this is meant as a resource this is acceptable.

Referee #1 (Remarks for Author):

Thank you for addressing my comments.

Referee #2 (Comments on Novelty/Model System for Author):

This is one of the first studies using spatial transcriptomics in human coronary plaques. The GeoMx and CosMx systems are used and give good insights into the spatial transcriptome of the plaque. Computational biology techniques are well explained, and highly innovative for his type of analysis.

Referee #2 (Remarks for Author):

The authors have improved their analysis. The paper is of high value for the atherosclerosis field. However, a biological follow-up paper is being awaited.

Referee #4 (Comments on Novelty/Model System for Author):

It is a very descriptive resource paper.

Referee #4 (Remarks for Author):

In their response letter, the authors stated that neighborhood and cell-cell interaction analyses are "outside the remit of this project." However, this position is not convincing, as such analyses are fundamental components of spatial transcriptomics studies. Furthermore, in the manuscript's Conclusion, the authors explicitly state: "We also constructed cell-cell interaction networks and conducted neighborhood analyses, which identified unique cellular clusters based on cell type composition and spatial coordinates." This directly acknowledges the significance of these analyses in the context of CosMx data. Given this, it is inconsistent to claim that cell-cell interaction and neighborhood analyses fall outside the scope of the manuscript.

Dear Lise,

We have carefully gone through the editorial issues and addressed them to the best of our abilities.

As per the issue raised by reviewer #4, we do agree that neighbourhood and cell-cell interaction analyses are of importance in these types of studies and hence we did perform them; however, we considered the further analyses requested out of those by this reviewer, ie whether there were any genes whose expression is influenced by cell-cell interactions, and whether immune cells exhibit neighbour-dependent gene expression via e.g. CellChat, CellNeighborEx, to be out of the scope of this study (lines 361-364).

As explained via email, source data for Fig 1A are too large and we have made them available in Pathcore at link <https://propath.pathcore.com/folder/6115?s=PXqg9GYLwBTp> where they can be downloaded. All other source data are appended as requested as zipped folders (one/figure).

We hope our revisions are satisfactory.

In faith,

Claudio Mauro (on behalf of all authors)

11th Jul 2025

Dear Prof. Mauro,

I am pleased to inform you that your manuscript is accepted for publication and is now being sent to our publisher to be included in the next available issue of EMBO Molecular Medicine!

With kind regards,

Lise
